# Emergence of brain-like mirror-symmetric viewpoint tuning in convolutional neural networks

**Amirhossein Farzmahdi[1,2], Wilbert Zarco[1], Winrich A Freiwald[1,3], Nikolaus Kriegeskorte[4,5,6,7], Tal Golan[4]***

[1]Laboratory of Neural Systems, The Rockefeller University, New York, United States; [2]School of Cognitive Sciences, Institute for Research in Fundamental Sciences, Tehran, Islamic Republic of Iran; [3]The Center for Brains, Minds & Machines, Cambridge, United States; [4]Zuckerman Mind Brain Behavior Institute, Columbia University, New York, United States; [5]Department of Psychology, Columbia University, New York, United States; [6]Department of Neuroscience, Columbia University, New York, United States; [7]Department of Electrical Engineering, Columbia University, New York, United States

***For correspondence:**
golan.neuro@bgu.ac.il

**Competing interest:** The authors declare that no competing interests exist.

**Abstract** Primates can recognize objects despite 3D geometric variations such as in-depth rotations. The computational mechanisms that give rise to such invariances are yet to be fully understood. A curious case of partial invariance occurs in the macaque face-patch AL and in fully connected layers of deep convolutional networks in which neurons respond similarly to mirror-symmetric views (e.g. left and right profiles). Why does this tuning develop? Here, we propose a simple learning-driven explanation for mirror-symmetric viewpoint tuning. We show that mirror-symmetric viewpoint tuning for faces emerges in the fully connected layers of convolutional deep neural networks trained on object recognition tasks, even when the training dataset does not include faces. First, using 3D objects rendered from multiple views as test stimuli, we demonstrate that mirror-symmetric viewpoint tuning in convolutional neural network models is not unique to faces: it emerges for multiple object categories with bilateral symmetry. Second, we show why this invariance emerges in the models. Learning to discriminate among bilaterally symmetric object categories induces reflection-equivariant intermediate representations. AL-like mirror-symmetric tuning is achieved when such equivariant responses are spatially pooled by downstream units with sufficiently large receptive fields. These results explain how mirror-symmetric viewpoint tuning can emerge in neural networks, providing a theory of how they might emerge in the primate brain. Our theory predicts that mirror-symmetric viewpoint tuning can emerge as a consequence of exposure to bilaterally symmetric objects beyond the category of faces, and that it can generalize beyond previously experienced object categories.

## Editor's evaluation

This computational study is a valuable empirical investigation into the common trait of neurons in brains and artificial neural networks: responding effectively to both objects and their mirror images and it focuses on uncovering conditions that lead to mirror symmetry in visual networks and the evidence convincingly demonstrates that learning contributes to expanding mirror symmetry tuning, given its presence in the data. Additionally, the paper delves into the transformation of face patches in primate visual hierarchy, shifting from view specificity to mirror symmetry to view invariance. It empirically analyzes factors behind similar effects in many network architectures, and key claims

highlight the emergence of invariances in architectures with spatial pooling, driven by learning bilateral symmetry discrimination and importantly, these effects extend beyond faces, suggesting broader relevance.

## Introduction

Primates can recognize objects robustly despite considerable image variation. Although we experience object recognition as immediate and effortless, the process involves a large portion of cortex and considerable metabolic cost (*Laughlin et al., 1998*), and determining the neural mechanisms and computational principles that enable this ability remains a major neuroscientific challenge. One particular object category, faces, offers an especially useful window into how the visual cortex transforms retinal signals to object representations. The macaque brain contains a network of interconnected areas devoted to the processing of faces. This network, the face-patch system, forms a subsystem of the inferotemporal (IT) cortex (*Tsao et al., 2006*; *Moeller et al., 2008*; *Freiwald and Tsao, 2010*; *Hesse and Tsao, 2020*). Neurons across the network show response selectivity for faces, but are organized in face patches–spatially and functionally distinct modules (*Freiwald and Tsao, 2010*; *Freiwald, 2020*). These patches exhibit an information processing hierarchy from posterior to anterior areas. In the most posterior face-patch, PL (posterior lateral), neurons respond to face components (*Issa and DiCarlo, 2012*). In ML/MF (middle lateral/middle fundus), neurons respond to whole faces in a view-specific manner. In AL (anterior lateral), responses are still view-specific, but mostly reflection-invariant. Finally in AM (anterior medial), neurons respond with sensitivity to the identity of the face, but in a view-invariant fashion (*Freiwald and Tsao, 2010*). The average neuronal response latencies increase across this particular sequence of stages (*Freiwald and Tsao, 2010*). Thus, it appears as if visual information is transformed across this hierarchy of representational stages in a way that facilitates the recognition of individual faces despite view variations.

What are the computational principles that give rise to the representational hierarchy evident in the face-patch system? Seeking potential answers to this and similar questions, neuroscientists have been increasingly turning to convolutional neural networks (CNNs) as baseline computational models of the primate ventral visual stream. Although CNNs lack essential features of the primate ventral stream, such as recurrent connectivity, they offer a simple hierarchical model of its feedforward cascade of linear-non-linear transformations. Feedforward CNNs remain among the best models for predicting mid- and high-level cortical representations of novel natural images within the first 100–200ms after stimulus onset (*Yamins et al., 2014*; *Khaligh-Razavi and Kriegeskorte, 2014*). Diverse CNN models, trained on tasks such as face identification (*Farzmahdi et al., 2016*; *Abudarham et al., 2021*; *Raman and Hosoya, 2020*), object recognition (*Chang et al., 2021*), inverse graphics (*Yildirim et al., 2020*), sparse coding (*Hosoya and Hyvärinen, 2017*), and unsupervised generative modeling *Higgins et al., 2021* have all been shown to replicate at least some aspects of face-patch system representations. Face-selective artificial neurons occur even in untrained CNNs (*Baek et al., 2021a*), and functional specialization between object and face representation emerges in CNNs trained on the dual task of recognizing objects and identifying faces (*Dobs et al., 2022*).

To better characterize and understand the computational mechanisms employed by the primate face-patch system and test whether the assumptions implemented by current CNN models are sufficient for explaining its function, we should carefully inspect the particular representational motifs the face-patch system exhibits. One of the more salient and intriguing of these representational motifs is the *mirror-symmetric viewpoint tuning* in the AL face-patch (*Freiwald and Tsao, 2010*). Neurons in this region typically respond with different firing rates to varying views of a face (e.g. a lateral profile vs. a frontal view), but they respond with similar firing rates to views that are horizontal reflections of each other (e.g. left and right lateral profiles) (*Freiwald and Tsao, 2010*).

To date, two distinct computational models have been put forward as potential explanations for AL's mirror-symmetric viewpoint tuning. Leibo and colleagues (*Leibo et al., 2017*) considered unsupervised learning in an HMAX-like (*Riesenhuber and Poggio, 1999*) four-layer neural network exposed to a sequence of face images rotating in depth about a vertical axis. When the learning of the mapping from the complex-cell-like representation of the second layer to the penultimate layer was governed by Hebbian-like synaptic updates (*Oja, 1982*), approximating a principal components analysis (PCA) of the input images, the penultimate layer developed mirror-symmetric viewpoint tuning.

In another modeling study, Yildirim and colleagues (*Yildirim et al., 2020*) trained a CNN to invert the rendering process of 3D faces, yielding a hierarchy of intermediate and high-level face representations. Mirror-symmetric viewpoint tuning emerged in an intermediate representation between two densely-connected transformations mapping 2.5D surface representations to high-level shape and texture face-space representations. Each of these two models (*Leibo et al., 2017*; *Yildirim et al., 2020*) provides a plausible explanation of AL's mirror-symmetric viewpoint tuning, but each requires particular assumptions about the architecture and learning conditions, raising the question whether a more general computational principle can provide a unifying account of the emergence of mirror-symmetric viewpoint tuning.

Here, we propose a parsimonious, bottom-up explanation for the emergence of mirror-symmetric viewpoint tuning for faces (*Figure 1*). We find that learning to discriminate among bilaterally symmetric object categories promotes the learning of representations that are *reflection-equivariant* (i.e. they code a mirror image by a mirrored representation). Spatial pooling of the features, as occurs in the transition between the convolutional and fully connected layers in CNNs, then yields *reflection-invariant* representations (i.e. these representations code a mirror image as they would code the original image). These reflection-invariant representations are not fully view-invariant: They are still tuned to particular views of faces (e.g. respond more to a half-profile than to a frontal view, or vice versa), but they do not discriminate between mirrored views. In other words, these representations exhibit mirror-symmetric viewpoint tuning (in the twin sense of the neuron responding equally to left-right-reflected images and the tuning function, hence, being mirror-symmetric). We propose that the same computational principles may explain the emergence of mirror-symmetric viewpoint tuning in the primate face-patch system.

Our results further suggest that emergent reflection-invariant representations may also exist for non-face objects: the same training conditions give rise to CNN units that show mirror-symmetric tuning profiles for non-face objects that have a bilaterally symmetric structure. Extrapolating from CNNs back to primate brains, we predict AL-like mirror-symmetric viewpoint tuning in non-face-specific visual regions that are parallel to AL in terms of the ventral stream representational hierarchy. Such tuning could be revealed by probing these regions with non-face objects that are bilaterally symmetric.

## Results

### Deep layers in CNNs exhibit mirror-symmetric viewpoint tuning to multiple object categories

We investigated whether reflection-invariant yet view-specific tuning emerges naturally in deep convolutional neural networks. To achieve this, we generated a diverse set of 3D objects rendered in multiple views. We evaluated the hidden-layer activations of an ImageNet-trained AlexNet CNN model (*Krizhevsky et al., 2012*) presented with nine views of each object exemplar. We constructed a $9 \times 9$ representational dissimilarity matrix (RDM; *Kriegeskorte et al., 2008*) for each exemplar object and each CNN layer, summarizing the view tuning of the layer's artificial neurons ('units') by means of between-view representational distances. The resulting RDMs revealed a progression throughout the CNN layers for objects with one or more symmetry planes: These objects induce mirror-symmetric RDMs in the deeper CNN layers (*Figure 2A*), reminiscent of the symmetric RDMs measured for face-related responses in the macaque AL face-patch (*Freiwald and Tsao, 2010*). We defined a 'mirror-symmetric viewpoint tuning index' to quantify the degree to which representations are view-selective yet reflection-invariant (*Figure 2B*). Consider a dissimilarity matrix $D \in \mathbb{R}^{n \times n}$ where $D_{j,k}$ denotes the distance between view $j$ and view $k$, $n$ denotes the number of views. The RDM is symmetric about the main diagonal by definition: $D_{j,k} = D_{k,j}$, independent of the tuning of the units. The views are ordered from left to right, such that $j$ and $n+1-k$ refer to horizontally reflected views. The mirror-symmetric viewpoint tuning index is defined as the Pearson linear correlation coefficient between $D$ and its horizontally flipped counterpart, $D^H_{j,k} = D_{j,n+1-k}$ (*Equation 1*). Note that this is equivalent to the correlation between vertically flipped RDMs, because of the symmetry of the RDMs about the diagonal: $D^H_{j,k} = D_{j,n+1-k} = D^V_{j,k} = D_{n+1-j,k}$. This mirror-symmetric viewpoint tuning index is positive and large to the extent that the units are view-selective but reflection-invariant (like the neurons in macaque AL

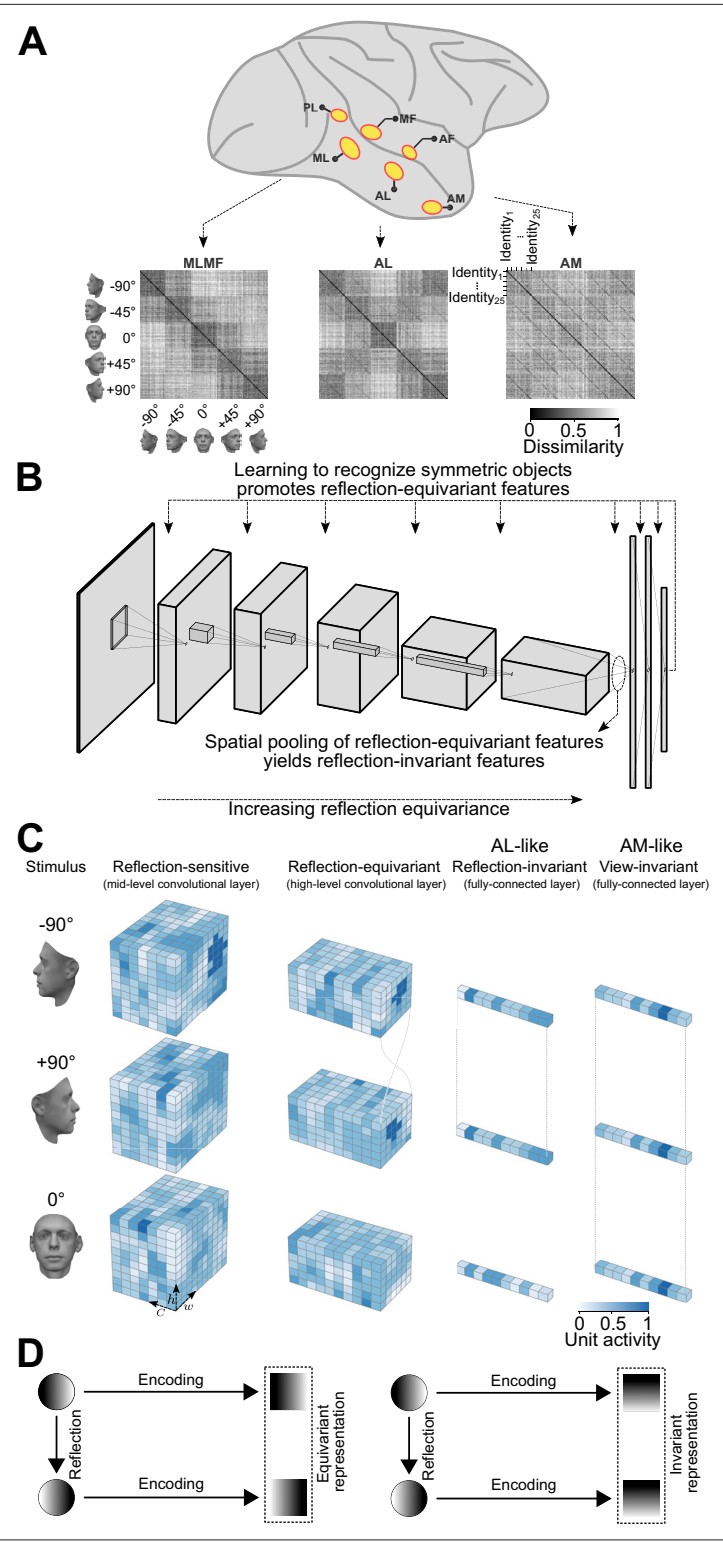

**Figure 1.** An overview of our claim: convolutional deep neural networks trained on discriminating among bilaterally symmetric object categories provide a parsimonious explanation for the mirror-symmetric viewpoint tuning of the macaque AL face-patch. (**A**) The macaque face-patch system. Face-selective cortical areas are highlighted in yellow. The areas ML, AL, and AM exhibit substantially different tuning propreties when presented with faces of different head orientations (***Freiwald and Tsao, 2010***). These distinct tuning profiles are evident in population-level representational dissimilarity matrices (RDMs). From posterior to anterior face

*Figure 1 continued on next page*

*Figure 1 continued*

areas, invariance to viewpoints gradually increases: from view-tuned in ML, through mirror-symmetric in AL, to view-invariant identity selectivity in AM (neural data from ***Freiwald and Tsao, 2010***). (**B**) Training convolutional deep neural networks on recognizing specific symmetric object categories (e.g. faces, cars, the digit 8) gives rise to AL-like mirror-symmetric tuning. It is due to a cascade of two effects: First, learning to discriminate among symmetric object categories promotes tuning for reflection-equivariant representations throughout the entire processing layers. This reflection equivariance increases with depth. Then, long-range spatial pooling (as in the transformation of the last convolution layer to the first fully connected layer in CNNs) transforms the equivariant representations into reflection-invariant representations. (**C**) Schematic representations of three viewpoints of a face (left profile, frontal view, right profile) are shown in three distinct stages of processing. Each tensor depicts the width (**w**), height (**h**), and depth (**c**) of an activation pattern. Colors indicate channel activity. From left to right: In a mid-level convolutional layer, representations are view-specific. A deeper convolutional layer produces reflection-equivariant representations that are view-specific. Feature vectors of a fully connected layer become invariant to reflection by pooling reflection-equivariant representations from the last convolutional layer. (**D**) A graphical comparison of reflection-equivariance and reflection-invariance. Circles denote input images, and squares denote representations.

---

face-patch). The index is near zero for units with view-invariant tuning (such as the AM face-patch), where the dissimilarities are all small and any variations are caused by noise.

***Figure 2C*** displays the average mirror-symmetric viewpoint tuning index for each object category across AlexNet layers. Several categories—faces, chairs, airplanes, tools, and animals—elicited low (below 0.1) or even negative mirror-symmetric viewpoint tuning values throughout the convolutional layers, transitioning to considerably higher (above 0.6) values starting from the first fully connected layer (fc6). In contrast, for fruits and flowers, mirror-symmetric viewpoint tuning was low in both the convolutional and the fully connected layers. For cars and boats, mirror-symmetric viewpoint tuning was notably high already in the shallowest convolutional layer and remained so across the network's layers. To explain these differences, we quantified the symmetry of the various 3D objects in each category by analyzing their 2D projections (***Figure 2—figure supplement 1***). We found that all of the categories that show high mirror-symmetric viewpoint tuning index in fully connected but not convolutional layers have a single plane of symmetry. For example, the left and right halves of a human face are reflected versions of each other (***Figure 2D***). This 3D structure yields symmetric 2D projections only when the object is viewed frontally, thus hindering lower level mirror-symmetric viewpoint tuning. Cars and boats have two planes of symmetry: in addition to the symmetry between their left and right halves, there is an approximate symmetry between their back and front halves. The quintessential example of such quadrilateral symmetry would be a Volkswagen Beetle viewed from the outside. Such 3D structure enables mirror-symmetric viewpoint tuning even for lower-level representations, such as those in the convolutional layers. Fruits and flowers exhibit radial symmetry but lack discernible symmetry planes, a characteristic that impedes viewpoint tuning altogether.

However, for an untrained AlexNet, the mirror-symmetric viewpoint tuning index remains relatively constant across the layers (***Figure 2—figure supplement 2A***). Statistically contrasting mirror-symmetric viewpoint tuning between a trained and untrained AlexNet demonstrates that the leap in mirror-symmetric viewpoint tuning in fc6 is training-dependent (***Figure 2—figure supplement 2B***).

Shallow and deep convolutional neural network models with varied architectures and objective functions replicate the emergence of mirror-symmetric viewpoint tuning (***Figure 2—figure supplement 3***). These models include VGG16 (***Simonyan and Zisserman, 2015***), 'VGGFace' network (trained on face identification) (***Parkhi et al., 2015***), EIG (***Yildirim et al., 2020***), HMAX (***Riesenhuber and Poggio, 1999***), ResNet50 (***He et al., 2016***), ConvNeXt (***Liu et al., 2022***). In all these convolutional networks, the mirror-symmetric viewpoint tuning index peaks at the fully-connected or average pooling layers. ViT (***Dosovitskiy et al., 2021***), featuring a non-convolutional architecture, does not exhibit this feature (***Figure 2—figure supplement 5***).

Why does the transition to the fully connected layers induce mirror-symmetric viewpoint tuning for bilaterally symmetric objects? One potential explanation is that the learned weights that map the last convolutional representation (pool5) to the first fully connected layer (fc6) combine the pool5 activations in a specific pattern that induces mirror-symmetric viewpoint tuning. However, replacing fc6 with spatial global average pooling (collapsing each pool5 feature map into a scalar activation) yields a representation with very similar mirror-symmetric viewpoint tuning levels (***Figure 2—figure***

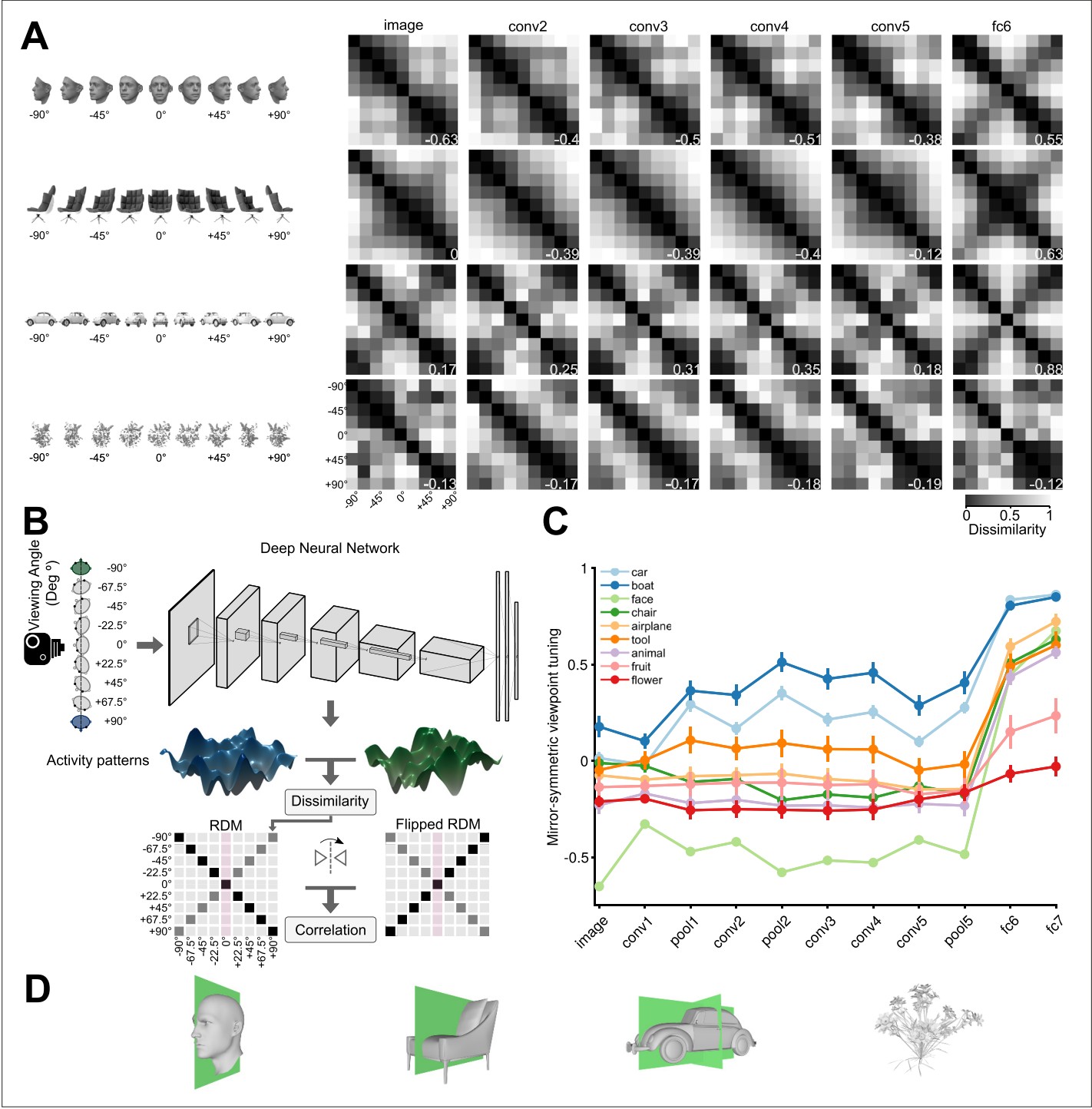

**Figure 2.** Mirror-symmetric viewpoint tuning of higher level deep neural network representations emerges for multiple object categories. (**A**) Different viewpoint tuning across the layers of AlexNet for four example objects. For each object, the responses to nine views (–90° to +90° in the steps of 22.5°) were measured in six key AlexNet layers, shallow (input, *left*) to deep (fc6, *right*). For each layer, a Representational Dissimilarity Matrix (RDM) depicts how the population activity vector varies across different object views. Each element of the RDM represents the dissimilarity (1 - Pearson correlation coefficient) between a pair of activity vectors evoked in response to two particular views. The symmetry of the RDMs about the major diagonal is inherent to their construction. However, the symmetry about the minor diagonal (for the face and chair, in fc6, and for the car, already in conv2) indicates mirror-symmetric viewpoint tuning. (**B**) The schematic shows how the mirror-symmetric viewpoint tuning index was quantified. We first fed the network with images of each object from nine viewpoints and recorded the activity patterns of its layers. Then, we computed the dissimilarity between activity patterns of different viewpoints to create an RDM. Next, we measured the correlation between the obtained RDM and its horizontally

*Figure 2 continued on next page*

*Figure 2 continued*

flipped counterpart, excluding the frontal view (which is unaffected by the reflection). (**C**) The Mirror-symmetric viewpoint tuning index across all AlexNet layers for nine object categories (car, boat, face, chair, airplane, animal, tool, fruit, and flower). Each solid circle denotes the average of the index over 25 exemplars within each object category. Error bars indicate the standard error of the mean. The mirror-symmetric viewpoint tuning index values of the four example objects in panel B are shown at the bottom right of each RDM in panel B. *Figure 2—figure supplement 4* shows the same analysis applied to representations of the face stimulus set used in *Freiwald and Tsao, 2010*, across various neural network models. (**D**) 3D Objects have different numbers of symmetry axes. A face (left column), a non-face object with bilateral symmetry (a chair, second column), an object with quadrilateral symmetry (a car, third column), and an object with no obvious reflective symmetry planes (a flower, right column).

The online version of this article includes the following figure supplement(s) for figure 2:

**Figure supplement 1.** Assessment of symmetry planes in 3D renders across viewpoints.

**Figure supplement 2.** The mirror-symmetric viewpoint tuning index remains unchanged as the signal moves into the fully connected layers of the untrained network.

**Figure supplement 3.** Convolutional networks, regardless of their architecture and training objectives, exhibit peak mirror-symmetric viewpoint tuning at the fully-connected and average pooling layers.

**Figure supplement 4.** Mirror-symmetric viewpoint tuning of various neural network architectures measured with respect to the FIV face stimulus set (*Freiwald and Tsao, 2010*) and compared to the mirror-symmetric viewpoint tuning of three face-patches (MLMF, AL, and AM).

**Figure supplement 5.** The highest mirror-symmetric viewpoint tuning index across all layers of each evaluated neural network model.

**Figure supplement 6.** One of the key operations in fully-connected layers is spatial pooling.

**Figure supplement 7.** Layer-wise mirror-symmetric viewpoint tuning profiles measured by linear correlation without employing unit-specific z-score normalization.

**Figure supplement 8.** Comparison of mirror-symmetric viewpoint tuning in a supervised, PCA-based model (*Leibo et al., 2017*) and a supervised CNN (AlexNet) trained on object recognition.

supplement 6). This result is suggestive of an alternative explanation: that training the network on ImageNet gives rise to a reflection-equivariant representation in pool5. We therefore investigated the reflection equivariance of the convolutional representations.

## Reflection equivariance versus reflection invariance of convolutional layers

Consider a representation $f(\cdot)$, defined as a function that maps input images to sets of feature maps, and a geometric image transformation $g(\cdot)$, applicable to either feature maps or raw images. $f$ is equivariant under $g$ if $f(g(x)) = g(f(x))$ for any input image $x$ (see also *Kvinge et al., 2022*). While convolutional feature maps are approximately equivariant under translation (but see *Azulay and Weiss, 2019*), they are not in general equivariant under reflection or rotation. For example, an asymmetrical filter along reflection axes in the first convolutional layer would yield an activation map that is not equivariant under reflection. And yet, the demands of the task on which a CNN is trained may lead to the emergence of representations that are approximately equivariant under reflection or rotation (see *Cohen and Welling, 2016*; *Weiler et al., 2018* for neural network architectures that are equivariant to reflection or rotation by construction). If a representation $f$ is equivariant under a transformation $g$ that is a spatial permutation of its input (e.g. $g$ is a horizontal or vertical reflection or a 90° rotation) then $f(x)$ and $f(g(x))$ are spatially permuted versions of each other. If a spatially invariant function $h(\cdot)$ (i.e. a function that treats the pixels as a set, such as the average or the maximum) is then applied to the feature maps, the composed function $h \circ f$ is *invariant* to $g$ since $h\big(f(g(x))\big) = h\big(g(f(x))\big) = h(f(x))$. Transforming a stack of feature maps into a channel vector by means of global average pooling is a simple case of such a spatially invariant function $h$. Therefore, if task-training induces approximately reflection-equivariant representations in the deepest convolutional layer of a CNN and approximately uniform pooling in the following fully connected layer, the resulting pooled representation would be approximately reflection-invariant.

We examined the emergence of approximate equivariance and invariance in CNN layers (*Figure 3*). We considered three geometric transformations: horizontal reflection, vertical reflection, and 90° rotation. Note that given their architecture alone, CNNs are not expected to show greater equivariance and invariance for horizontal reflection compared to vertical reflection or 90° rotation. However, greater invariance and equivariance for horizontal reflection may be expected on the basis of natural image statistics and the demands of invariant recognition. Many object categories in the natural world

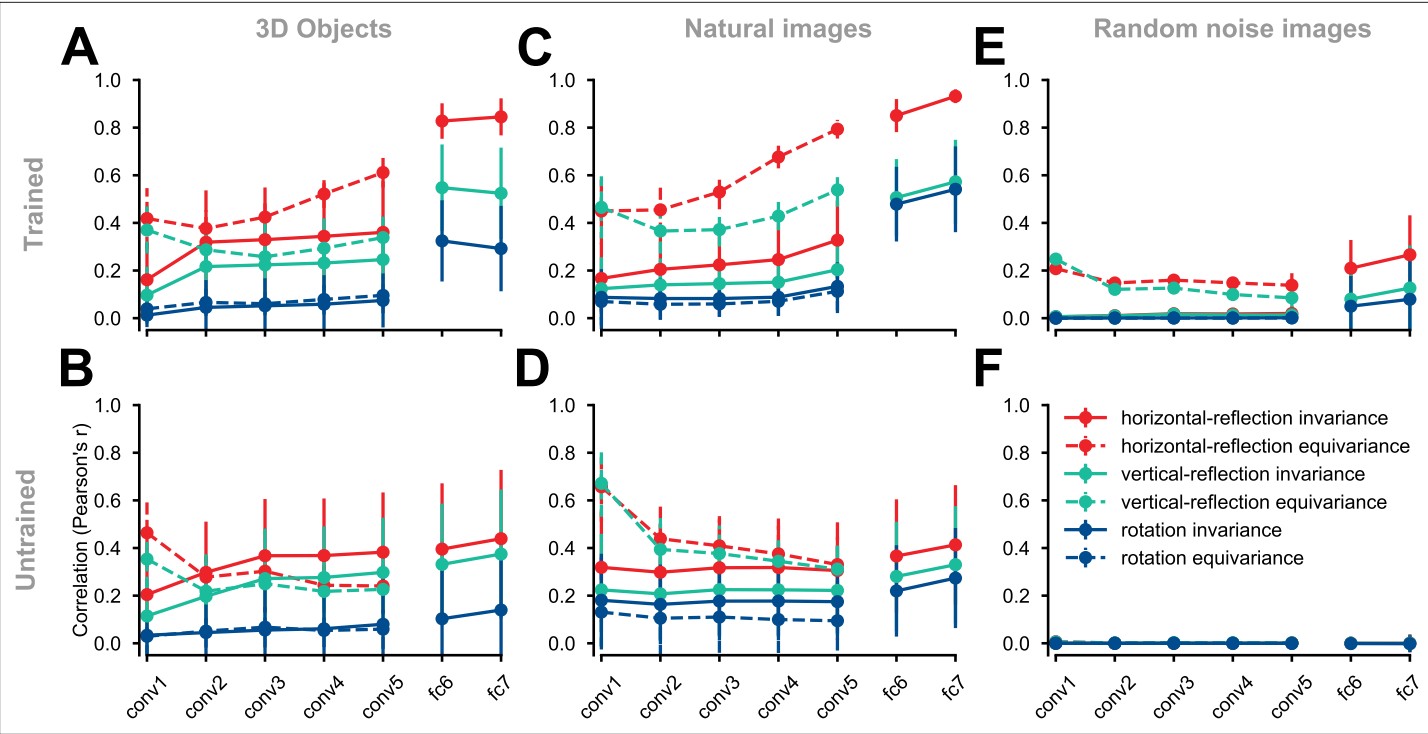

**Figure 3.** Equivariance and invariance in trained and untrained deep convolutional neural networks. Each solid circle represents an equivariance or invariance measure, averaged across images. Hues denote different transformations (horizontal flipping, vertical flipping, or 90° rotation). Error bars depict the standard deviation across images (each test condition consists of 2025 images). Invariance is a measure of similarity between the activity pattern an image elicits and the activity pattern its transformed (e.g. flipped) counterpart (solid lines) elicits. Equivariance is a measure of the similarity between the activity pattern of a transformed image elicits and the *transformed* version of the activity pattern the untransformed image elicits (dashed lines). In the convolutional layers, both invariance and equivariance can be measured. In the fully connected layers, whose representations have no explicit spatial structure, only invariance is measurable. (**A**) ImageNet-trained AlexNet tested on the rendered 3D objects. (**B**) Untrained AlexNet tested on rendered 3D objects. (**C**) ImageNet-trained AlexNet tested on the natural images (images randomly selected from the test set of ImageNet). (**D**) Untrained AlexNet tested on the natural images. (**E**) ImageNet-trained AlexNet tested on the random noise images. (**F**) Untrained AlexNet tested on the random noise images.

The online version of this article includes the following figure supplement(s) for figure 3:

**Figure supplement 1.** Image-specific representational invariance and equivariance across 3D object renders, natural images, and random noise images, measured in a deep convolutional neural network (AlexNet) trained on ImageNet or alternatively, left untrained.

**Figure supplement 2.** Training-induced enhancement of horizontal reflection invariance in the first fully connected layer (fc6), across different object categories.

are bilaterally symmetric with respect to a plane parallel to the axis of gravity and are typically viewed (or photographed) in an upright orientation. Horizontal image reflection, thus, tends to yield equally natural images of similar semantic content, whereas vertical reflection and 90° rotation yield unnatural images.

To measure equivariance and invariance, we presented the CNNs with pairs of original and transformed images. To measure the invariance of a fully-connected CNN layer, we calculated an across-unit Pearson correlation coefficient for each pair of activation vectors that were induced by a given image and its transformed version. We averaged the resulting correlation coefficients across all image pairs (Materials and methods, *Equation 2*). For convolutional layers, this measure was applied after flattening stacks of convolutional maps into vectors. In the case of horizontal reflection, this invariance measure would equal 1.0 if the activation vectors induced by each image and its mirrored version are identical (or perfectly correlated).

Equivariance could be quantified only in convolutional layers because units in fully connected layers do not form visuotopic maps that can undergo the same transformations as images. It was quantified similarly to invariance, except that we applied the transformation of interest (i.e. reflection or rotation) not only to the image but also to the convolutional map of activity elicited by the untransformed

image (*Equation 3*). We correlated the representation of the transformed image with the transformed representation of the image. In the case of horizontal reflection, this equivariance measure would equal 1.0 if each activation map induced by an image and its reflected version are reflected versions of each other (or are perfectly correlated after horizontally flipping one of them).

We first evaluated equivariance and invariance with respect to the set of 3D object images described in the previous section. In an ImageNet-trained AlexNet, horizontal-reflection equivariance increased across convolutional layers (*Figure 3A*). Equivariance under vertical reflection was less pronounced and equivariance under 90° rotation was even weaker (*Figure 3A*). In this trained AlexNet, invariance jumped from a low level in convolutional layers to a high level in the fully connected layers and was highest for horizontal reflection, lower for vertical reflection, and lowest for 90° rotation.

In an untrained AlexNet, the reflection equivariance of the first convolutional layer was higher than in the trained network. However, this measure subsequently decreased in the deeper convolutional layers to a level lower than that observed for the corresponding layers in the trained network. The higher level of reflection-equivariance of the first layer of the untrained network can be explained by the lack of strongly oriented filters in the randomly initialized layer weights. While the training leads to oriented filters in the first layer, it also promotes downstream convolutional representations that have greater reflection-equivariance than those in a randomly-initialized, untrained network.

The gap between horizontal reflection and vertical reflection in terms of both equivariance and invariance was less pronounced in the untrained network (*Figure 3B*), indicating a contribution of task training to the special status of horizontal reflection. In contrast, the gap between vertical reflection and 90° rotation in terms of both equivariance and invariance was preserved in the untrained network. This indicates that the greater degree of invariance and equivariance for vertical reflection compared to 90° rotation is largely caused by the test images' structure rather than task training. One interpretation is that, unlike 90° rotation, vertical and horizontal reflection both preserve the relative prevalence of vertical and horizontal edge energy, which may not be equal in natural images (*Coppola et al., 1998*; *Torralba and Oliva, 2003*; *Henderson and Serences, 2021*; *Girshick et al., 2011*). To test if the emergence of equivariance and invariance under horizontal reflection is unique to our controlled stimulus set (which contained many horizontally symmetrical images), we repeated these analyses using natural images sampled from the ImageNet validation set (*Figure 3C–D*). The training-dependent layer-by-layer increase in equivariance and invariance to horizontal reflection was as pronounced for natural images as it was for the rendered 3D object images. Therefore, the emergent invariance and equivariance under horizontal reflection are not an artifact of the synthetic object stimulus set.

Repeating these analyses on random noise images, the ImageNet-trained AlexNet still showed a slightly higher level of horizontal reflection-equivariance (*Figure 3E*), demonstrating the properties of the features learned in the task independently of symmetry structure in the test images. When we evaluated an untrained AlexNet on random noise images (*Figure 3F*), that is, when there was no structure in either the test stimuli or the network weights, the differences between horizontal reflection, vertical reflection, and rotation measures disappeared, and the invariance and equivariance measures were zero, as expected (see *Figure 3—figure supplement 1* for the distribution of equivariance and invariance across test images and *Figure 3—figure supplement 2* for analysis of horizontal reflection invariance across different object categories).

To summarize this set of analyses, a high level of reflection-invariance is associated with the layer's pooling size and the reflection-equivariance of its feeding representation. The pooling size depends only on the architecture, but the reflection-equivariance of the feeding representation depends on both architecture and training. Training on recognizing objects in natural images induces a greater degree of invariance and equivariance to horizontal reflection compared to vertical reflection or 90° rotation. This is consistent with the statistics of natural images as experienced by an upright observer looking, along a horizontal axis, at upright bilaterally symmetric objects. Image reflection, in such a world ordered by gravity, does not change the category of an object (although rare examples of dependence of meaning on handedness exist, such as the letters p and q, and molecules whose properties depend on their chirality). However, the analyses reported thus far leave unclear whether natural image statistics alone or the need to disregard the handedness for categorization drive mirror-symmetric viewpoint tuning. In the following section, we examine what it is about the training that drives viewpoint tuning to be mirror-symmetric.

## Learning to discriminate among categories of bilaterally symmetric objects induces mirror-symmetric viewpoint tuning

To examine how task demand and visual diet influence mirror-symmetric viewpoint tuning, we trained four deep convolutional neural networks of the same architecture on different datasets and tasks (*Figure 4*). The network architecture and training hyper-parameters are described in the Materials and Methods section (for training-related metrics, see *Figure 4—figure supplement 1*). Once trained, each network was evaluated on the 3D object images used in *Figure 2*, measuring mirror-symmetric viewpoint tuning qualitatively (*Figure 4B*) and quantitatively (*Figure 4C*).

First, we considered a network trained on CIFAR-10 (*Krizhevsky and Hinton, 2009*), a dataset of small images of 10 bilaterally symmetric categories (airplanes, cars, birds, cats, deer, dogs, frogs, horses, ships, and trucks). Although this dataset contains no human face images (such images appear coincidentally in the ImageNet dataset, *Yang et al., 2022*), the CIFAR-10-trained network reproduced the result of a considerable level of mirror-symmetric viewpoint tuning for faces in layers fc1 and fc2 (*Figure 4B*, top row). This network also showed mirror-symmetric viewpoint tuning for other bilaterally symmetric objects such as cars, airplanes, and boats (*Figure 4C*, blue lines).

We then considered a network trained on SVHN (Street View House Numbers; *Netzer et al., 2011*), a dataset of photographs of numerical digits. Its categories are mostly asymmetric (since all 10 digits except for '0' and '8' are asymmetric). Unlike the network trained on CIFAR-10, the SVHN-trained network showed a very low level of mirror-symmetric viewpoint tuning for faces. Furthermore, its levels of mirror-symmetric viewpoint tuning for cars, airplanes, and boats were reduced relative to the CIFAR-10-trained network.

SVHN differs from CIFAR-10 both in its artificial content and the asymmetry of its categories. To disentangle these two factors, we designed a modified dataset, 'symSVHN'. Half of the images in symSVHN were horizontally reflected SVHN images. All the images maintained their original category labels (e.g. images of the digit 7 and images of a mirrored 7 belonged to the same category). We found that the symSVHN-trained network reproduced the mirror-symmetric viewpoint tuning observed in the CIFAR-10-trained network.

Last, we modified the labels of symSVHN such that the flipped digits would count as 10 separate categories, in addition to the 10 unflipped digit categories. This dataset ('asymSVHN') has the same images as symSVHN, but it is designed to require reflection-sensitive recognition. The asymSVHN-trained network reproduced the low levels of mirror-symmetric viewpoint tuning observed for the original SVHN dataset. Together, these results suggest that given the spatial pooling carried out by fc1, the task demand of *reflection-invariant recognition* is a sufficient condition for the emergence of mirror-symmetric viewpoint tuning for faces.

## Equivariant local features drive mirror-symmetric viewpoint tuning

What are the image-level visual features that drive the observed mirror-symmetric viewpoint tuning? Do mirror-reflected views of an object induce similar representations because of global 2D configurations shared between such views? Or alternatively, are reflection-equivariant local features sufficient to explain the finding of similar responses to reflected views in fc1? We used a masking-based importance mapping technique (*Petsiuk et al., 2018*) to characterize which features drive the responses of units with mirror-symmetric viewpoint tuning. First, we created importance maps whose elements represent how local features influence each unit's response to different object views. The top rows of panels A and B in *Figure 5* show examples of such maps for two units, one that shows considerable mirror-symmetric viewpoint tuning for cars and another that shows considerable mirror-symmetric viewpoint tuning for faces.

Next, we empirically tested whether the local features highlighted by the importance maps are sufficient and necessary for generating mirror-symmetric viewpoint tuning. We used two image manipulations: insertion and deletion (*Petsiuk et al., 2018*; *Figure 5A–B*, middle rows). When we retained only the most salient pixels (i.e. insertion), we observed that the units' mirror-symmetric viewpoint tuning levels were similar to those induced by unmodified images (*Figure 5A–B*, dark blue lines). This result demonstrates that the local features suffice for driving mirror-symmetrically tuned responses. Conversely, greying out the most salient pixels (deletion) led to a complete loss of mirror-symmetric viewpoint tuning (*Figure 5A–B*, red lines). This result demonstrates that the local features are necessary to drive mirror-symmetrically tuned responses. To examine this effect systematically,

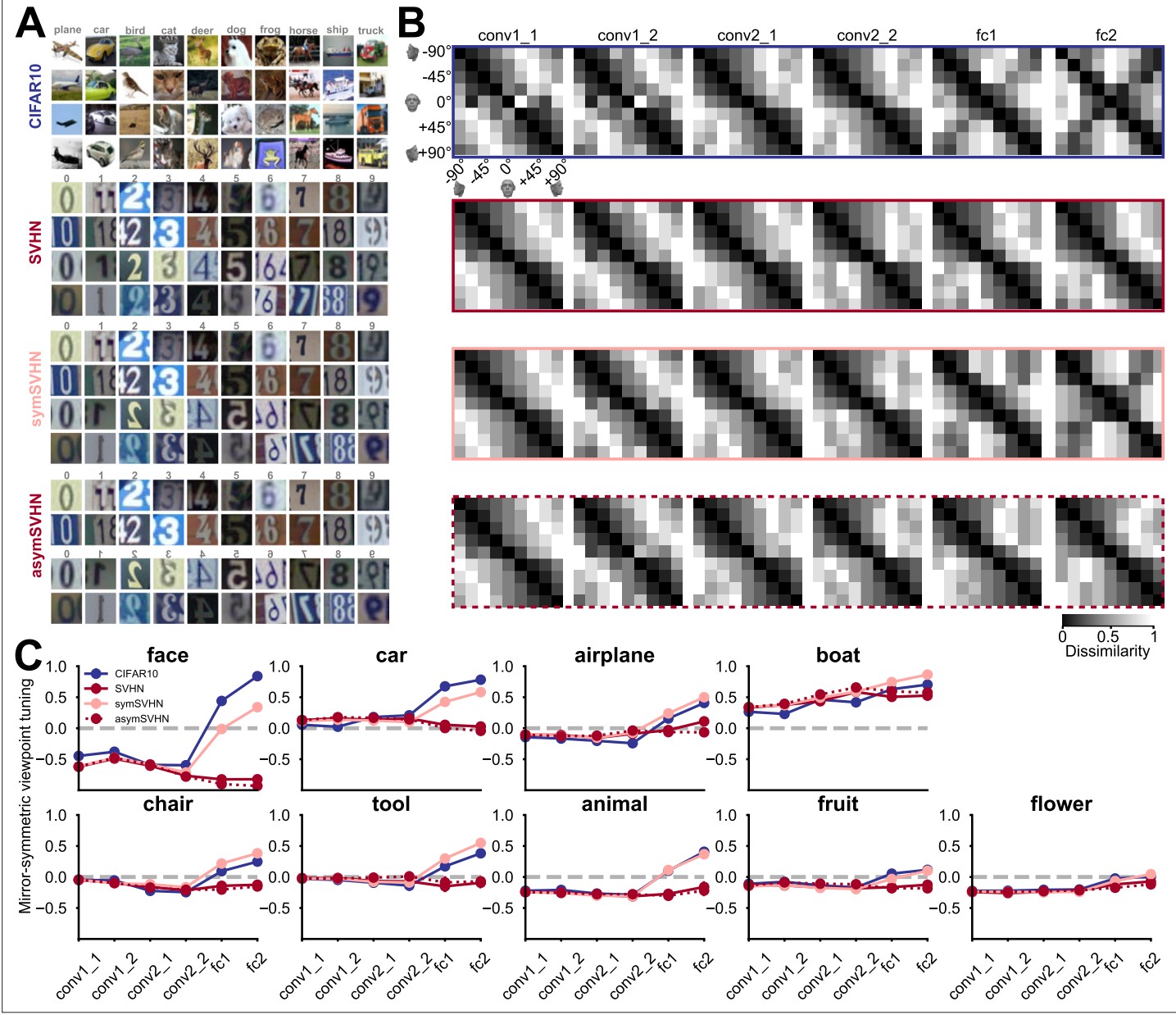

**Figure 4.** The effect of training task and training dataset on mirror-symmetric viewpoint tuning. (**A**) Four datasets are used to train deep neural networks of the same architecture: CIFAR-10, a natural image dataset with ten bilaterally symmetric object categories; SVHN, a dataset with mostly asymmetric categories (the ten numerical digits); symSVHN, a version of the SVHN dataset in which the categories were made bilaterally symmetric by horizontally reflecting half of the training images (so the digit 7 and its mirrored version count as members of the same category); asymSVHN, the same image set as in symSVHN but with the mirrored images assigned to ten new distinct categories (so the digit 7 and its mirrored version count as members of distinct categories). (**B**) Each row represents the RDMs of the face exemplar images from nine viewpoints for each trained network corresponding to its left side panel. Each entry of the RDM represents the dissimilarity (1 - Pearson's r) between two pairs of image-induced activity vectors in the corresponding layer. The RDMs' order from left to right refers to the depth of layers within the network. As the dissimilarity color bar indicates, the dissimilarity values increase from black to white color. (**C**) Mirror-symmetric viewpoint tuning index values across layers for nine object categories in each of the four networks. The solid circles refer to the average of the index across 25 exemplars within each object category for three networks trained on 10 labels. The red dashed line with open circles belongs to the asymSVHN network trained on 20 labels. The gray dashed lines indicate the index of zero. Error bars represent the standard error of the mean calculated across exemplars.

The online version of this article includes the following figure supplement(s) for figure 4:

**Figure supplement 1.** Network learning curves.

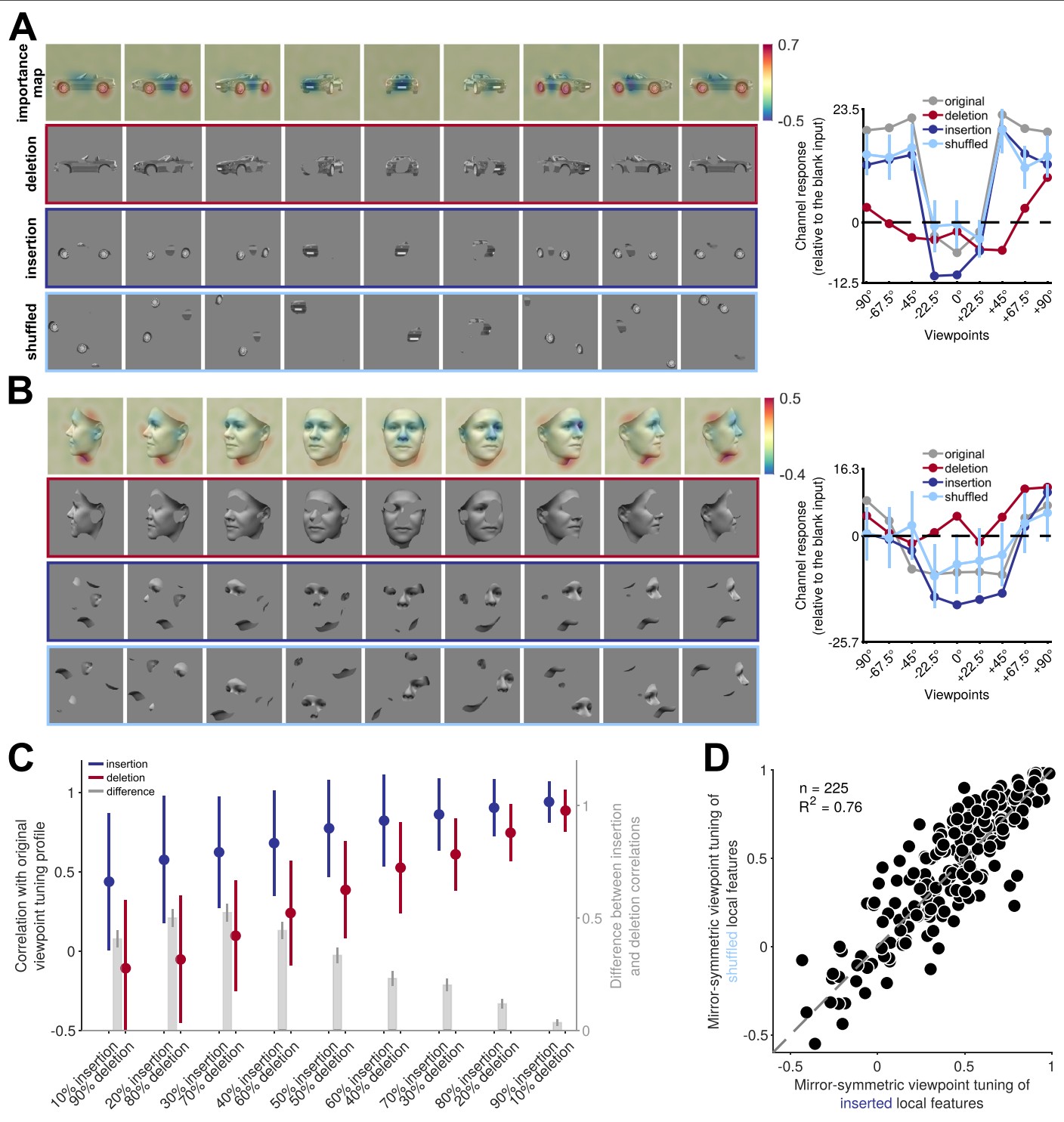

**Figure 5.** Reflection-invariant viewpoint-specific responses are driven mostly by local features. This figure traces image-level causes for the mirror-symmetric viewpoint tuning using Randomized Input Sampling for Explanation (RISE, *Petsiuk et al., 2018*). (**A**) Analysis of the features of different views of a car exemplar that drive one particular unit in fully connected layer fc6 of AlexNet. The topmost row in each panel depicts an image-specific *importance map* overlaid to each view of the car, charting the contribution of each pixel to the unit's response. The second row ('deletion') depicts a version of each input image in which the 25% most contributing pixels are masked with the background gray color. The third row ('insertion') depicts a version of the input images in which only the most contributing 25% of pixels appear. The last row represents the shuffled spatial configuration of extracted local features, which maintains their structure and changes their locations. The charts on the right depict the units' responses to the original, deletion, insertion, and shuffled images. The dashed line indicates the units' response to a blank image. The y-axis denotes the unit's responses

*Figure 5 continued on next page*

*Figure 5 continued*

compared to its response to a blank image. (**B**) Analogous analysis of the features of different views of a face that drive a different unit in fully connected layer fc6 of AlexNet. (**C**) Testing local contributions to mirror-symmetric viewpoint tuning across all object exemplars and insertion/deletion thresholds. For each object exemplar, we selected a unit with a highly view-dependent but symmetric viewpoint tuning (the unit whose tuning function was maximally correlated with its reflection). We then measured the correlation between this tuning function and the tuning function induced by insertion or deletion images that were generated by a range of thresholding levels (from 10 to 90%). Note that each threshold level consists of images with the same number of non-masked pixels appearing in the insertion and deletion conditions. In the insertion condition, only the most salient pixels are retained, and in the deletion condition, only the least salient pixels are retained. The solid circles and error bars indicate the median and standard deviation over 225 objects, respectively. The right y-axis depicts the difference between insertion and deletion conditions. Error bars represent the SEM. (**D**) For each of 225 objects, we selected units with mirror-symmetric viewpoint tuning above the 95 percentile (≈200 units) and averaged their corresponding importance maps. Next, we extracted the top 25% most contributing pixels from the averaged maps (insertion) and shuffled their spatial configuration (shuffled). We then measured the viewpoint-RDMs for either the inserted or shuffled object image set. The scatterplot compares the mirror-symmetric viewpoint tuning index between insertion and shuffled conditions, calculated across the selected units. Each solid circle represents an exemplar object. The high explained variance indicates that the global configuration does not play a significant role in the emergence of mirror-symmetric viewpoint tuning.

The online version of this article includes the following figure supplement(s) for figure 5:

**Figure supplement 1.** The emergence of mirror symmetric weight tensors in AlexNet.

**Figure supplement 2.** Individual neural network units exhibiting mirror-symmetric view tuning according to the criterion employed by *Baek et al., 2021a*.

**Figure supplement 3.** Selecting individual units with genuine mirror-symmetric viewpoint tuning.

**Figure supplement 4.** Training-dependent emergence of units with mirror-symmetric viewpoint tuning across neural network layers.

we selected one unit for each of the 225 3D objects that showed high mirror-symmetric viewpoint tuning. We then tested these 225 units with insertion and deletion images produced with different thresholds (*Figure 5C*). Across all threshold levels, the response to insertion images was more similar to the response to unmodified images, whereas deletion images failed to induce mirror-symmetric viewpoint tuning.

These results indicate a role for local features in mirror-symmetric tuning. However, the features may form larger-scale configurations synergistically. To test the potential role of such configurations, we shuffled contiguous pixel patches that were retained in the insertion condition. This manipulation destroyed global structure while preserving local features (*Figure 5A–B*, bottom row). We found that the shuffled images largely preserved the units' mirror-symmetric viewpoint tuning (*Figure 5D*). Thus, it is the mere presence of a similar set of reflected local features (rather than a reflected global configuration) that explains most of the acquired mirror-symmetric viewpoint tuning. Note that such local features must be either symmetric at the image level (e.g. the wheel of a car in a side view), or induce a reflection-equivariant representation (e.g. an activation map that highlights profile views of a nose, regardless of their orientation). The fc6 layer learns highly symmetrical weight maps, reducing the sensitivity to local feature configurations and enabling the generation of downstream reflection-invariant representations compared to convolutional layers (*Figure 5—figure supplement 1*).

## Representational alignment between artificial networks and macaque face patches

How does the emergence of mirror-invariance in CNNs manifest in the alignment of these networks with neural representations of faces in the macaque face-patch system? In line with (*Yildirim et al., 2020*), we reanalyzed the neural recordings from *Freiwald and Tsao, 2010* by correlating neural population RDMs, each describing the dissimilarities among neural responses to face images of varying identities and viewpoints, with corresponding model RDMs, derived from neural network layer representations of the stimulus set (*Figure 6*, top row). In addition to the AL face-patch, we considered MLMF, which is sensitive to reflection (*Freiwald and Tsao, 2010*), and AM, which is mostly viewpoint invariant (*Freiwald and Tsao, 2010*). Following the approach of Yildirim and colleagues, the neural networks were presented with segmented reconstructions, where non-facial pixels were replaced by a uniform background.

Consistent with previous findings (*Yildirim et al., 2020*), MLMF was more aligned with the CNNs' mid-level representation, notably the last convolutional layers (*Figure 6A*). The AL face patch showed its highest representational alignment with the first fully connected layer (*Figure 6B*), coinciding with

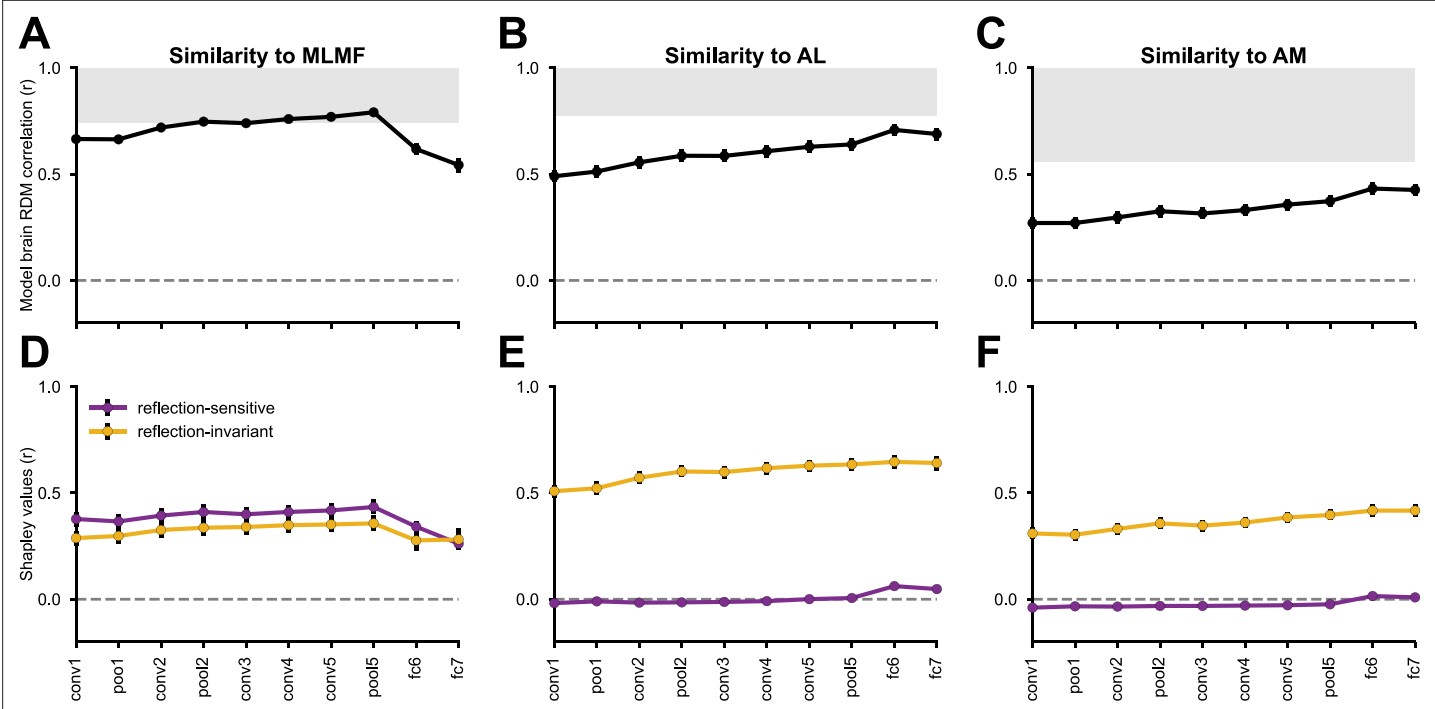

**Figure 6.** Reflection-invariant and reflection-sensitive contributions to the representational similarity between monkey face patch neurons and AlexNet layers. The neural responses were obtained from *Freiwald and Tsao, 2010*, where electrophysiological recordings were conducted in three faces patches while the monkeys were presented with human faces of various identities and views. (Top row) linear correlations between RDMs from each network layer and each monkey face patch (MLMF, AL, AM). Error bars represent standard deviations estimated by bootstrapping individual stimuli (see Materials and methods). The gray area represents the neural data's noise ceiling, whose lower bound was determined by Spearman-Brown-corrected split-half reliability, with the splits applied across neurons. (Bottom row) Each model–brain RDM correlation is decomposed into the additive contribution of two feature components: reflection-sensitive (purple) and reflection-invariant (yellow). *Figure 6—figure supplements 1–3* present the same analyses applied to a diverse set of neural network models, across the three regions.

The online version of this article includes the following figure supplement(s) for figure 6:

**Figure supplement 1.** Alignment of MLMF and neural network representations across diverse architecures.

**Figure supplement 2.** Alignment of AL and neural network representations across diverse architecures.

**Figure supplement 3.** Alignment of AM and neural network representations across diverse architecures.

the surge of the mirror-symmetric viewpoint tuning index at this processing level (see *Figure 2*). The AM face patch aligned most with the fully connected layers (*Figure 6C*).

These correlations between model and neural RDMs reflect the contribution of multiple underlying image features. To disentangle the contribution of reflection-invariant and reflection-sensitive representations to the resulting RDM correlation, we computed two additional model representations for each neural network layer: (1) a reflection-invariant representation, obtained by element-wise addition of two activation tensors, one elicited in response to the original stimuli and the other in response to mirror-reflected versions of the stimuli; and, (2) a reflection-sensitive representation, obtained by element-wise subtraction of these two tensors. The two resulting feature components sum to the original activation tensor; a fully reflection-invariant representation would be entirely accounted for by the first component. For each CNN layer, we obtained the two components and correlated each of them with the unaltered neural RDMs. Through the Shapley value feature attribution method (*Shapley, 1953*), we transformed the resulting correlation coefficients into additive contributions of the reflection-invariant and reflection-sensitive components to the original model-brain RDM correlations (*Figure 6D–F*).

In the MLMF face patch, reflection-sensitive features contributed more than reflection-invariant ones, consistent with the dominance of reflection-sensitive information in aligning network layers with MLMF data (*Figure 6D*). Conversely, in the AL and AM face patches, reflection-invariant features accounted for nearly all the observed model–brain RDM correlations (*Figure 6E, F*). For most of the

convolutional layers, the contribution of the reflection-sensitive component to AL or AM alignment was negative—meaning that if the layers' representations were more reflection-invariant, they could have explained the neural data better.

## Discussion

In this paper, we propose a simple learning-driven explanation for the mirror-symmetric viewpoint tuning for faces in the macaque AL face-patch. We found that CNNs trained on object recognition reproduce this tuning in their fully connected layers. Based on in silico experiments, we suggest two jointly sufficient conditions for the emergence of mirror-symmetric viewpoint tuning. First, training the network to discriminate among bilaterally symmetric 3D objects yields reflection-equivariant representations in the deeper convolutional layers. Then, subsequent pooling of these reflection-equivariant responses by units with large receptive fields leads to reflection-invariant representations with mirror-symmetric view tuning similar to that observed in the AL face patch. Like our models, monkeys need to recognize bilaterally symmetric objects that are oriented by gravity. To achieve robustness to view, the primate visual system can pool responses from earlier stages of representation. We further show that in CNNs, such tuning is not limited to faces and occurs for multiple object categories with bilateral symmetry. This result yields a testable prediction for primate electrophysiology and fMRI.

### Mirror-symmetric viewpoint tuning in brains and machines

Several species, including humans, confuse lateral mirror images (e.g. the letters b and d) more often than vertical mirror images (e.g. the letters b and p; *Sutherland, 1960*; *Todrin and Blough, 1983*). Children often experience this confusion when learning to read and write (*Nelson and Peoples, 1975*; *Bornstein et al., 1978*; *Cornell, 1985*; *Dehaene et al., 2010*). Single-cell recordings in macaque monkeys presented with simple stimuli indicate a certain degree of reflection-invariance in IT neurons (*Rollenhagen and Olson, 2000*; *Baylis and Driver, 2001*). Human neuroimaging experiments also revealed reflection-invariance across higher-level visual regions for human heads (*Axelrod and Yovel, 2012*; *Kietzmann et al., 2012*; *Ramírez et al., 2014*; *Kietzmann et al., 2017*) and other bilaterally symmetric objects (*Dilks et al., 2011*; *Ramírez et al., 2014*).

When a neuron's response is reflection-invariant and yet the neuron responds differently to different object views, it is exhibiting mirror-symmetric viewpoint tuning. Such tuning has been reported in a small subset of monkeys' STS and IT cells in early recordings (*Perrett et al., 1991*; *Logothetis et al., 1995*). fMRI-guided single-cell recordings revealed the prevalence of this tuning profile among the cells of face patch AL (*Freiwald and Tsao, 2010*). The question of why mirror-symmetric viewpoint tuning emerges in the cortex has drawn both mechanistic and functional explanations. Mechanistic explanations suggest that mirror-symmetric viewpoint tuning is a by-product of increasing interhemispheric connectivity and receptive field sizes. Due to the anatomical symmetry of the nervous system and its cross-hemispheric interconnectivity, mirror-image pairs activate linked neurons in both hemispheres (*Corballis and Beale, 1976*; *Gross et al., 1977*). A functional perspective explains partial invariance as a stepping stone toward achieving fully view-invariant object recognition (*Freiwald and Tsao, 2010*). Our results support a role for both of these explanations. We showed that global spatial pooling is a sufficient condition for the emergence of reflection-invariant responses, *if* the pooled representation is reflection-equivariant. Global average pooling extends the spatially integrated stimulus region. Likewise, interhemispheric connectivity may result in cells with larger receptive fields that cover both hemifields.

A recent work by *Revsine et al., 2024* incorporated biological constraints, including interhemispheric connectivity, into a model processing solely low-level stimulus features, namely intensity and contrast. Their results suggest that such features might be sufficient for explaining apparent mirror-symmetric viewpoint tuning in fMRI studies. In our study, we standardized stimulus intensity and contrast across objects and viewpoints (see Methods), eliminating these features as potential confounds. Additionally, applying a dissimilarity measure that is invariant to the overall magnitude of the representations did not alter the observed trends in mirror-symmetric viewpoint tuning results (*Figure 2—figure supplement 7*). Therefore, we suggest that spatial pooling can yield genuine mirror-symmetric viewpoint tuning in CNNs and brains by summating equivariant mid-level visual features (see *Figure 5*) that are learning-dependent (*Figure 4*).

We also showed that equivariance can be driven by the task demand of discriminating among objects that have bilateral symmetry (see *Olah et al., 2020* for an exploration of emergent equivariance using activation maximization). The combined effect of equivariance and pooling leads to a leap in reflection-invariance between the last convolutional layer and the fully connected layers in CNNs. This transition may be similar to the transition from view-selective cells in face patches ML/MF to mirror-symmetric viewpoint-selective cells in AL. In both CNNs and primate cortex, the mirror-symmetrically viewpoint-tuned neurons are a penultimate stage on the path to full view invariance (*Freiwald and Tsao, 2010*).

## Unifying the computational explanations of mirror-symmetric viewpoint tuning

Two computational models have been suggested to explain AL's mirror-symmetric viewpoint tuning, the first attributing it to Hebbian learning with Oja's rule (*Leibo et al., 2017*), the second to training a CNN to invert a face-generative model (*Yildirim et al., 2020*). A certain extent of mirror-symmetric viewpoint tuning was also observed in CNNs trained on face identification (Fig. 3Eii in *Yildirim et al., 2020*, Figure 2 in *Raman and Hosoya, 2020*). In light of our findings here, these models can be viewed as special cases of a considerably more general class of models. Our results generalize the computational account in terms of both stimulus domain and model architecture. Both (*Leibo et al., 2017*) and (*Yildirim et al., 2020*) trained neural networks with face images. Here, we show that it is not necessary to train on a specific object category (including faces) in order to acquire reflection equivariance and invariance for exemplars of that category. Instead, learning mirror-invariant stimulus-to-response mappings gives rise to equivariant and invariant representations also for novel stimulus classes.

Our claim that mirror-symmetric viewpoint tuning is learning-dependent may seem to be in conflict with findings by Baek and colleagues (*Baek et al., 2021a*). Their work demonstrated that units with mirror-symmetric viewpoint tuning profile can emerge in randomly initialized networks. Reproducing Baek and colleagues' analysis, we confirmed that such units occur in untrained networks (*Figure 5—figure supplement 3*). However, we also identified that the original criterion for mirror-symmetric viewpoint tuning employed in *Baek et al., 2021a* was satisfied by many units with asymmetric tuning profiles (*Figure 5—figure supplements 2 and 3*). Once we applied a stricter criterion, we observed a more than twofold increase in mirror-symmetric units in the first fully connected layer of a trained network compared to untrained networks of the same architecture (*Figure 5—figure supplement 4*). This finding highlights the critical role of training in the emergence of mirror-symmetric viewpoint tuning in neural networks also at the level of individual units.

Our results also generalize the computational account of mirror-symmetric viewpoint tuning in terms of the model architectures. The two previous models incorporated the architectural property of spatial pooling: the inner product of inputs and synaptic weights in the penultimate layer of the HMAX-like model in *Leibo et al., 2017* and the global spatial pooling in the f4 layer of the EIG model (*Yildirim et al., 2020*). We showed that in addition to the task, such spatial pooling is an essential step toward the emergence of mirror-symmetric tuning in our findings.

## Limitations

The main limitation of the current study is that our findings are simulation-based and empirical in nature. Therefore, they might be limited to the particular design choices shared across the range of CNNs we evaluated. This limitation stands in contrast with the theoretical model proposed by Leibo and colleagues (*Leibo et al., 2017*), which is reflection-invariant by construction. However, it is worth noting that the model proposed by Leibo and colleagues is reflection-invariant only with respect to the horizontal center of the input image (*Figure 2—figure supplement 8*). CNNs trained to discriminate among bilaterally symmetric categories develop mirror-symmetric viewpoint tuning across the visual field (*Figure 2—figure supplement 8*). The latter result pattern is more consistent with the relatively position-invariant response properties of AL neurons (Fig. S10 in *Freiwald and Tsao, 2010*).

A second consequence of the simulation-based nature of this study is that our findings only establish that mirror-symmetric viewpoint tuning is a viable computational means for achieving view invariance; they do not prove it to be a necessary condition. In fact, previous modeling studies *Farzmahdi et al., 2016*; *Leibo et al., 2015*; *Leibo et al., 2017* have demonstrated that a direct transition from

view-specific processing to view invariance is possible. However, in practice, we observe that both CNNs and the face-patch network adopt solutions that include intermediate representations with mirror-symmetric viewpoint tuning.

## A novel prediction: mirror-symmetric viewpoint tuning for non-face objects

Mirror-symmetric viewpoint tuning has been mostly investigated using face images. Extrapolating from the results in CNNs, we hypothesize that mirror-symmetric viewpoint tuning for non-face objects should exist in cortical regions homologous to AL. The mirror-symmetric tuning of these objects does not necessarily have to be previously experienced by the animal.

This hypothesis is consistent with the recent findings of *Bao et al., 2020*. They report a functional clustering of IT into four separate networks. Each of these networks is elongated across the IT cortex and consists of three stages of processing. We hypothesize that the intermediate nodes of the three non-face selective networks have reflection-invariant yet view-selective tuning, analogous to AL's representation of faces.

Our controlled stimulus set, which includes systematic 2D snapshots of 3D real-world naturalistic objects, is available online. Future electrophysiological and fMRI experiments utilizing this stimulus set can verify whether the mirror-symmetric viewpoint tuning for non-face categories we observe in task-trained CNNs also occurs in the primate IT.

## Methods

### 3D object stimulus set

We generated a diverse image set of 3D objects rendered from multiple views in the depth rotation. Human faces were generated using the Basel Face Model (*Gerig et al., 2018*). For the non-face objects, we purchased access to 3D models on TurboSquid (http://www.turbosquid.com). The combined object set consisted of nine categories (cars, boats, faces, chairs, airplanes, animals, tools, fruits, and flowers). Each category included 25 exemplars. We rendered each exemplar from nine views, giving rise a total of 2025 images. The views span from –90° (left profile) to +90°, with steps of 22.5°. The rendered images were converted to grayscale, placed on a uniform gray background, and scaled to 227 × 227 pixels to match the input image size of AlexNet, or to 224 × 224 to match the input image size of the VGG-like network architectures. Mean luminance and contrast of non-background pixels were equalized across images using the SHINE toolbox (*Willenbockel et al., 2010*).

### Pre-trained neural networks

We selected both shallow and deep networks with varied architectures and objective functions. We evaluated convolutional networks trained on ImageNet, including AlexNet (*Krizhevsky et al., 2012*), VGG16 (*Simonyan and Zisserman, 2015*), ResNet50, ConvNeXt. Additionally, we evaluated VGGFace–a similar architecture to VGG16, trained on the VGG Face dataset (*Parkhi et al., 2015*), ViT with its non-convolutional architecture, EIG as a face generative model, and the shallow, biologically inspired HMAX model. All these networks, except for VGGFace, EIG, and HMAX, were trained on the ImageNet dataset (*Russakovsky et al., 2015*), which consists of ~1.2 million natural images from 1000 object categories (available on Matlab Deep Learning Toolbox and Pytorch frameworks, *The MathWorks Inc, 2019*; *Paszke et al., 2019*). The VGGFace model was trained on ~2.6 million face images from 2622 identities (available on the MatConvNet library, *Vedaldi and Lenc, 2015*). Each convolutional network features a distinct number of convolutional (conv), max-pooling (pool), rectified linear unit (relu), normalization (norm), average pooling (avgpool), and fully connected (fc) layers, among others, dictated by its architecture. For untrained AlexNet and VGG16 networks, we initialized the weights and biases using a random Gaussian distribution with a zero mean and a variance inversely proportional to the number of inputs per unit (*LeCun et al., 2012*).

### Trained-from-scratch neural networks

To control for the effects of the training task and 'visual diet', we trained four networks employing the same convolutional architecture on four different datasets: CIFAR-10, SVHN, symSVHN, and asymSVHN.

## CIFAR-10

CIFAR-10 consists of 60,000 RGB images of 10 classes (airplane, automobile, bird, cat, deer, dog, frog, horse, ship, truck) downscaled to 32 × 32 pixels (*Krizhevsky and Hinton, 2009*). We randomly split CIFAR-10's designated training set into 45,000 images used for training and 5,000 images used for validation. No data augmentation was employed. The reported classification accuracy (*Figure 4— figure supplement 1*) was evaluated on the remaining 10,000 CIFAR-10 test images.

## SVHN

SVHN (*Netzer et al., 2011*) contains 99,289 RGB images of 10 digits (0–9) taken from real-world house number photographs (*Netzer et al., 2011*), cropped to character bounding boxes and downsized to 32 × 32 pixels. We split the dataset into 73,257 images for the training set and 26,032 images for the test set. As with the CIFAR-10 dataset, we randomly selected 10% of training images as the validation set.

## symSVHN and asymSVHN

As a control experiment, we horizontally flipped half of the SVHN training images while keeping their labels unchanged. This manipulation encouraged the model trained on these images to become reflection-invariant in its decisions. This dataset was labeled as 'symSVHN'. In a converse manipulation, we applied the same horizontal flipping but set the flipped images' labels to 10 new classes. Therefore, each image in this dataset pertained to one of 20 classes. This manipulation removed the shared response mapping of mirror-reflected images and encouraged the model trained on these images to become sensitive to the reflection operation. This dataset was labeled as 'asymSVHN'.

## Common architecture and training procedure

The networks' architecture resembled the VGG architecture. It contained two convolutional layers followed by a max-pooling layer, two additional convolutional layers, and three fully connected layers. The size of convolutional filters was set to 3 × 3 with a stride of 1. The four convolutional layers consisted of 32, 32, 64, and 128 filters, respectively. The size of the max-pooling window was set to 2 × 2 with a stride of 2. The fully-connected layers had 128, 256, and 10 channels and were followed by a softmax operation (the asymSVHN network had 20 channels in its last fully connected layer instead of 10). We added a batch normalization layer after the first and the third convolutional layers and a dropout layer (probability = 0.5) after each fully-connected layer to promote quick convergence and avoid overfitting.

The networks' weights and biases were initialized randomly using the uniform He initialization (*He et al., 2015*). We trained the models using 250 epochs and a batch size of 256 images. The CIFAR-10 network was trained using stochastic gradient descent (SGD) optimizer starting with a learning rate of $10^{-3}$ and momentum of 0.9. The learning rate was halved every 20 epochs. The SVHN/symSVHN/asymSVHN networks were trained using the Adam optimizer. The initial learning rate was set to $10^{-5}$ and reduced by half every 50 epochs. The hyper-parameters were determined using the validation data. The models reached around 83% test accuracy (CIFAR-10: 81%, SVHN: 89%, symSVHN: 83%, asymSVHN: 80%). *Figure 4—figure supplement 1* shows the models' learning curves.

## Measuring representational dissimilarities

For the analyses described in *Figures 2–4*, Deep layers in CNNs exhibit mirror-symmetric viewpoint tuning to multiple object categories, and Reflection equivariance versus reflection invariance of convolutional layers, we first normalized the activation level of each individual neural network unit by subtracting its mean response level across all images of the evaluated dataset and dividing it by its standard deviation. The dissimilarity between the representations of two stimuli in a particular neural network layer (*Figures 2 and 4*) was quantified as one minus the Pearson linear correlation coefficient calculated across all of the layer's units (i.e. across the flattened normalized activation vectors). The *similarity* between representations (*Figure 3*) was quantified by the linear correlation coefficient itself.

## Measuring mirror-symmetric viewpoint tuning

Using the representational dissimilarity measure described above, we generated an $n \times n$ dissimilarity matrix for each exemplar object $i$ and layer $\ell$, where $n$ is the number of views (9 in our dataset). Each element of the matrix, $D^i_{j,k}$, denotes the representational distance between views $j$ and $k$ of object exemplar $i$. The views are ordered such that $j$ and $n+1-k$ refer to horizontally reflected views. We measured the mirror-symmetric viewpoint tuning index of the resulting RDMs by

$$r_{msvt} = \frac{1}{N} \sum_{i=1}^{N} r(D^i, D^{iH}),$$ (1)

where $r(\cdot, \cdot)$ is the Pearson linear correlation coefficient across view pairs, $D^H$ refers to horizontally flipped matrix such that $D^H_{j,k} = D_{j,n+1-k}$, and $N$ refers to number of object exemplars. The frontal view (which is unaltered by reflection) was excluded from this measure to avoid spurious inflation of the correlation coefficient. Previous work quantified mirror-symmetric viewpoint tuning by comparing neural RDMs to idealized mirror-symmetric RDM (see Fig. 3Ciii in *Yildirim et al., 2020*). Although highly interpretable, such an idealized RDM inevitably encompasses implicit assumptions about representational geometry that are unrelated to mirror-symmetry. For example, consider a representation featuring perfect mirror-symmetric viewpoint tuning and wherein for each view, the representational distances among all of the exemplars are equal. Its neural RDM would fit an idealized mirror-symmetric RDM better than the neural RDM of a representation featuring perfect mirror-symmetric viewpoint tuning yet non-equidistant exemplars. In contrast, the measure proposed in *Equation 1* equals 1.0 in both cases.

## Measuring equivariance and invariance

Representational equivariance and invariance were measured for an ImageNet-trained AlexNet and an untrained AlexNet with respect to three datasets: the 3D object image dataset described above, a random sample of 2025 ImageNet test images, and a sample of 2025 random noise images (*Figure 3*). Separately for each layer $\ell$ and image set $x_1, \ldots, x_{2025}$, we measured invariance by

$$r_{invariance} = \frac{1}{N} \sum_{i=1}^{N} r(f_\ell(x_i), f_\ell(g(x_i))),$$ (2)

where $f_\ell(\cdot)$ is the mapping from an input image $x$ to unit activations in layer $\ell$, $g(\cdot)$ is the image transformation of interest–vertical reflection, horizontal reflection, or rotation and $r$ is the Pearson linear correlation coefficient calculated across units, flattening the units' normalized activations into a vector in the case of convolutional layers. In order to estimate equivariance, we used the following definition:

$$r_{equivariance} = \frac{1}{N} \sum_{i=1}^{N} r(f_\ell(g(x_i)), g(f_\ell(x_i)))$$ (3)

Note that in this case, $g(\cdot)$ was applied both to the input images and the feature maps. This measure can be viewed as the inverse of an additive realization of latent space G-empirical equivariance deviation (G-EED; *Kvinge et al., 2022*). To prevent spurious correlations that may result from flipping and rotating operations, we have removed the central column when flipping horizontally, the central row when flipping vertically, and the central pixel when rotating 90 degrees. As a result, any correlations we observe are unbiased.

## Importance mapping

We used an established masking-based importance mapping procedure (*Petsiuk et al., 2018*) to identify visual features that drive units that exhibit mirror-symmetric viewpoint tuning profiles. Given an object for which the target unit showed mirror-symmetric viewpoint tuning, we dimmed the intensities of the images' pixels in random combinations to estimate the importance of image features. Specifically, for each image, we generated 5000 random binary masks. Multiplying the image with these masks yielded 5000 images in which different subsets of pixels were grayed out. These images were then fed to the network as inputs. The resulting importance maps are averages of these masks, weighted by target unit activity. To evaluate the explanatory power of the importance map of each

stimulus, we sorted the pixels according to their absolute values in the importance map and identified the top quartile of salient pixels. We then either retained ('insertion') or grayed out ('deletion') these pixels, and the resulting stimulus was fed into the network (*Figure 5A–B*). Due to the uniform gray background, we only considered foreground pixels. A second analysis compared viewpoint tuning between original images, deletion images, and insertion images across 10 thresholds, from 10% to 90%, with steps of 10% (*Figure 5C*). We conducted an additional analysis to examine the influence of global structure on the mirror-symmetric viewpoint tuning of the first fully connected layer (*Figure 5D*). To conduct this analysis at the unit population level, we generated one insertion image-set per object. First, we correlated each unit's view tuning curve against a V-shaped tuning template (i.e. a response proportional to the absolute angle of deviation from a frontal view) and retained only the units with positive correlations. We then correlated each unit's view-tuning curve with its reflected counterpart. We selected the top 5% most mirror-symmetric units (i.e. those showing the highest correlation coefficients).

For each object view, we generated an importance map for each of the selected units and averaged these maps across units. Using this average importance map, we generated an insertion image by retaining the top 25% most salient pixels. To test the role of global configuration, we generated a shuffled version of each insertion image by randomly relocating connected components. To assess model response to these images for each object exemplar, we computed the corresponding (9 × 9 views) RDM of fc1 responses given either the insertion images or their shuffled versions and quantified the mirror-symmetric viewpoint tuning of each RDM.

### Measuring brain alignment

To measure the alignment between artificial networks and macaque face patches, we used the face-identities-view (FIV) stimulus set (*Freiwald and Tsao, 2010*), as well as single-unit responses to these stimuli previously recorded from macaque face patches (*Freiwald and Tsao, 2010*). The FIV stimulus set includes images of 25 identities, each depicted in five views: left-profile, left-half profile, straight (frontal), right-half profile, and right-profile. The original recordings also included views of the head from upward, downward, and rear angles; these views were not analyzed in the current study to maintain comparability with its other analyses, which focused on yaw rotations. We measured the dissimilarity between the representations of each image pair using 1 minus the Pearson correlation and constructed an RDM. To assess the variability of this measurement, we adopted a stimulus-level bootstrap analysis, as outlined in *Yildirim et al., 2020*. A bootstrap sample was generated by selecting images with replacement from the FIV image set. From this sample, we calculated both the neural and model RDMs. To prevent spurious positive correlations, any nondiagonal identity pairs resulting from the resampling were removed. Subsequently, we determined the Pearson correlation coefficient between each pair of RDMs. This entire process was repeated across 1000 bootstrap samples. The authors declare no competing interest. The stimulus set and the source code required for reproducing our results are available at the following link: https://github.com/amirfarzmahdi/AL-Symmetry, (copy archived at *Farzmahdi, 2024*).

### Acknowledgements

Research reported in this publication was supported by the National Eye Institute of the National Institutes of Health under Award Numbers R01EY021594 and R01EY029998; by the National Institute Of Neurological Disorders And Stroke of the National Institutes of Health under Award Number RF1NS128897; and by the Department of the Navy, Office of Naval Research under ONR award number N00014-20-1-2292. This publication was made possible in part with the support of the Charles H Revson Foundation to TG. The content is solely the responsibility of the authors and does not necessarily represent the official views of the National Institutes of Health or the Charles H Revson Foundation. We thank Fernando Ramírez for an insightful discussion of an earlier version of this manuscript. We acknowledge Dr. T Vetter, Department of Computer Science, and the University of Basel, for the Basel Face Model.

## Additional information

### Funding

| Funder | Grant reference number | Author |
|---|---|---|
| National Eye Institute | R01EY021594 | Winrich A Freiwald |
| National Eye Institute | R01EY029998 | Winrich A Freiwald |
| National Institute of Neurological Disorders and Stroke | RF1NS128897 | Nikolaus Kriegeskorte |
| Naval Research Laboratory | N00014-20-1-2292 | Winrich A Freiwald |
| Charles H. Revson Foundation | | Tal Golan |

The funders had no role in study design, data collection and interpretation, or the decision to submit the work for publication.

### Author contributions

Amirhossein Farzmahdi, Conceptualization, Formal analysis, Visualization, Methodology, Writing – original draft, Writing – review and editing; Wilbert Zarco, Nikolaus Kriegeskorte, Conceptualization, Writing – review and editing; Winrich A Freiwald, Conceptualization, Resources, Writing – review and editing; Tal Golan, Conceptualization, Writing – original draft, Writing – review and editing

### Author ORCIDs

Amirhossein Farzmahdi  https://orcid.org/0000-0001-6926-546X
Wilbert Zarco  http://orcid.org/0000-0002-3599-0476
Winrich A Freiwald  https://orcid.org/0000-0001-8456-5030
Nikolaus Kriegeskorte  http://orcid.org/0000-0001-7433-9005
Tal Golan  http://orcid.org/0000-0002-7940-7473

### Decision letter and Author response

Decision letter https://doi.org/10.7554/eLife.90256.sa1
Author response https://doi.org/10.7554/eLife.90256.sa2

---

## Additional files

### Supplementary files

• MDAR checklist

### Data availability

The stimulus set and the source code required for reproducing our results are available at https://github.com/amirfarzmahdi/AL-Symmetry (copy archived at *Farzmahdi, 2024*).

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
