## [Editor Report]

This computational study is a valuable empirical investigation into the common trait of neurons in brains and artificial neural networks: responding effectively to both objects and their mirror images and it focuses on uncovering conditions that lead to mirror symmetry in visual networks and the evidence convincingly demonstrates that learning contributes to expanding mirror symmetry tuning, given its presence in the data. Additionally, the paper delves into the transformation of face patches in primate visual hierarchy, shifting from view specificity to mirror symmetry to view invariance. It empirically analyzes factors behind similar effects in many network architectures, and key claims highlight the emergence of invariances in architectures with spatial pooling, driven by learning bilateral symmetry discrimination and importantly, these effects extend beyond faces, suggesting broader relevance.

---

## [Decision Letter]

**Decision letter after peer review:**

Thank you for submitting your article "Emergence of brain-like mirror-symmetric viewpoint tuning in convolutional neural networks" for consideration by *eLife*. Your article has been reviewed by 3 peer reviewers, and the evaluation has been overseen by Mackenzie Mathis as Reviewing Editor and Yanchao Bi as the Senior Editor.

Essential revisions (for the authors):

Overall we collectively found the study valuable, but requires two major revisions to significantly strengthen the work. The concerns, and thus recommendations for major revision, largely revolve around the combination of choosing two weaker architectures and not establishing any correspondence to brain data. Thus, we agree to the two following essential revisions:

1. Please broaden the architectures that are being analyzed, including at least one state-of-the-art model for the effect you are studying. We think EIG is the most natural choice. To really strengthen the paper, recommend to include ResNets (strong computer vision models and good alignment to brain data), Transformers (strong computer vision models with different building blocks which would "stress-test" their learning and max-pooling claims), and HMAX (Leibo et al. claim this explains mirror-symmetric tuning, but not sure how well it aligns to data relative to EIG and other models).

2. Please defend your current choice of architectures by testing the brain alignment of VGG/AlexNet versus the current state-of-the-art EIG. we suggest performing comparisons of model activations to brain data for recordings where the subjects viewed mirror-symmetric stimuli (e.g., neural predictivity and/or CKA), or metrics that focus more explicitly on the mirror-symmetry effects, similar to Figure 3 in the Yildirim et al. paper. The most relevant result for connecting the models under study in this paper to neural recordings is this the paper by Yildirim et al. where VGG is found to be a poorer model than their proposed EIG. Depending on the choice of metric I think it is well possible that VGG is comparable to EIG, but it's up to the authors of this manuscript to defend their choice, and to make the comparison.

*Reviewer #1 (Recommendations for the authors):*

1. The potential mechanisms underlying mirror-symmetric tuning have been investigated in the past as has been acknowledged and intensively discussed in the paper. However, some of the conclusions made in this paper were already made in previous work, although not explicitly tested. As stated by Leibo et al. "the Hebb-type learning rule generates mirror-symmetric tuning for bilaterally symmetric objects, like faces", suggesting that it will also allow for mirror-symmetric tuning for other object categories. It would be important to acknowledge this. Moreover, while Yildirim et al. reports mirror-symmetric viewpoint tuning in VGGFace, they find that this tuning is less present than the view-invariant identity tuning, thereby contradicting neural findings in area AL. It would be beneficial to compare mirror-symmetry with view-invariant coding (as introduced in Figure 1A) in the fully-connected layers of the CNNs in this work, or at least to discuss and acknowledge this potential mismatch of CNNs with neural data.

2. The authors propose a mirror-symmetric viewpoint tuning index, which, although innovative, complicates comparison with previous work. This index is based on correlating representational dissimilarity matrices (RDMs) with their flipped versions, a method differing from previous approaches that correlated RDMs with idealized model RDMs corresponding to mirror symmetry. The mirror-symmetric tuning reported by Yildirim et al. for VGG-raw (corresponding to VGGFace in this study; Figure S2) appears significantly lower than the current study (Figure S1). To clarify these differences, an analysis using an idealized model, as in prior studies, or to better motivate the index used in this study would be valuable.

3. The analysis of reflection equivariance vs. invariance in both trained and untrained CNNs' convolutional layers is informative. However, an illustrative figure would enhance clarity and underscore the distinction between equivariance and invariance.

4. Faces seem to behave quite differently from all other object categories tested in this study. The mirror-symmetric viewpoint tuning for faces starts very low but then achieves similar values as other categories in the fully-connected layers. Moreover, the difference between task training and between trained and untrained AlexNet is maximal for faces (Figure 4 and S2). None of these differences are discussed in the paper. Given the critical role of faces for this study, the authors should provide speculations or potential explanations for these observations.

5. The study's analysis of training task and dataset raises some questions. Some categories, such as faces, seem to depend strongly on the task/dataset compared to others (e.g. boats). Acknowledging this finding and discussing potential reasons could enhance the manuscript.

6. It is surprising that the face-trained VGGFace shows lower mirror-symmetric viewpoint tuning as VGG16 (Figure S1). What could be possible explanations for this result? Could it be due to an overfitting to natural images of faces (compared to the Basel Face model faces used in this study) of the VGGFace network? A discussion of this finding would be beneficial to the paper.

7. The authors suggest that mirror-symmetric viewpoint tuning is a stepping stone towards view invariance. However, the study lacks explicit testing that this tuning is necessary for viewpoint invariance to emerge. For instance, if one prohibits a network to learn mirror-symmetric viewpoint tuning, would it still achieve viewpoint invariance for bilaterally symmetric objects? While such an analysis might be beyond the scope of this study, it could be beneficial to discuss this as a future direction.

*Reviewer #2 (Recommendations for the authors):*

I believe you have all the tools available to address the major weakness with respect to model architectures: https://github.com/CNCLgithub/EIG-faces provides code for the EIG model from Yildirim et al. 2020, and this model also includes pooling and versions of it have been trained for other datasets. As far as I am aware, the EIG model is state-of-the-art for the neural data of mirror-symmetric viewpoint tuning, so applying your analyses to this model would make them a lot more relevant and general.

You could also see if more recent "standard" convents such as ResNet-50 explain the neural data at least as well as EIG and then generalize your analyses to this (and other) model(s). To make this case you would have to quantitatively compare the two models on their neural alignment (I believe the data from Yildirim's paper should be public). You might also say that AlexNet and VGG are in the same ballpark for explaining the neural data, but then you should include that analysis in the paper and in the Yildirim paper, VGG seems to fall behind EIG's neural similarity by 10-20 percent points (Figure 3 D iv and E iv).

*Reviewer #3 (Recommendations for the authors):*

– It would be reassuring to know that the object classes have independent measures of symmetry *on which the networks operate*. If the statements about object-class-specific symmetry came after performing the experiments, then I recommend re-writing so that those statements are interpretations.

– It would be a great contribution to the field if the authors could clarify the relationship between this work and Baek et al.'s. Can they confirm that untrained networks have mirror-tuned units? If that is not replicable, then the "emergence" framing is accurate, and one can exercise appropriate weighing of that other study. This would help the field. If learning just amplifies this symmetry (by strengthening connections of view-dependent units, for example), that is also helpful to learn.

---

## [Author Response]

Essential revisions (for the authors):Overall we collectively found the study valuable, but requires two major revisions to significantly strengthen the work. The concerns, and thus recommendations for major revision, largely revolve around the combination of choosing two weaker architectures and not establishing any correspondence to brain data. Thus, we agree to the two following essential revisions:1. Please broaden the architectures that are being analyzed, including at least one state-of-the-art model for the effect you are studying. We think EIG is the most natural choice. To really strengthen the paper, recommend to include ResNets (strong computer vision models and good alignment to brain data), Transformers (strong computer vision models with different building blocks which would "stress-test" their learning and max-pooling claims), and HMAX (Leibo et al. claim this explains mirror-symmetric tuning, but not sure how well it aligns to data relative to EIG and other models).

We have extended our analyses to include EIG, ResNet50, ConvNeXt, ViT, and HMAX. The results reveal a common characteristic among convolutional networks that incorporate a fully-connected or average pooling layer, irrespective of their architecture (i.e., HMAX, EIG, ResNet50, and ConvNeXt). In all these networks, the mirror-symmetric viewpoint tuning index peaks around the fully-connected or average pooling layers. This result extends the generality of our previous observations in shallower networks such as VGG and AlexNet (see Figures 2—figure supplement 3 and 5).

2. Please defend your current choice of architectures by testing the brain alignment of VGG/AlexNet versus the current state-of-the-art EIG. we suggest performing comparisons of model activations to brain data for recordings where the subjects viewed mirror-symmetric stimuli (e.g., neural predictivity and/or CKA), or metrics that focus more explicitly on the mirror-symmetry effects, similar to Figure 3 in the Yildirim et al. paper. The most relevant result for connecting the models under study in this paper to neural recordings is this the paper by Yildirim et al. where VGG is found to be a poorer model than their proposed EIG. Depending on the choice of metric I think it is well possible that VGG is comparable to EIG, but it's up to the authors of this manuscript to defend their choice, and to make the comparison.

We evaluated the representational alignment of various architectures—AlexNet, VGG, EIG, ResNet50, ConvNeXt, and ViT against the neural dataset from Freiwald & Tsao 2010 (including three face patches), following the analytic conventions introduced by Yildirim and colleagues. Going beyond standard RSA, we disentangled the contribution of reflection invariant and reflection-sensitive features to the observed alignment (see Figure 6 and Figure 6—figure supplement 1-3). Last, we estimated the mirror-symmetric viewpoint tuning in the three face patches recorded by Freiwald & Tsao, allowing a direct comparison of this index in artificial neural networks and neural recordings (Figure 2—figure supplement 4).

In the revised manuscript, we no longer emphasize shallower convolutional architectures such as AlexNet or VGG16 as better models; the explanation convolutional neural networks offer to mirror-symmetric viewpoint tuning as observed in the face patch system applies also when using deeper models.

Manuscript changes: (main text, Results section)

“Representational alignment between artificial networks and macaque face patches

How does the emergence of mirror-invariance in CNNs manifest in the alignment of these networkswith neural representations of faces in the macaque face-patch system? In line with Yildirim andcolleagues (2020) [14], we reanalyzed the neural recordings from Freiwald and Tsao (2010) [4] by correlating neural population RDMs, each describing the dissimilarities among neural responses to face images of varying identities and viewpoints, with corresponding model RDMs, derived from neural network layer representations of the stimulus set (Figure 6, top row). In addition to the AL face-patch, we considered MLMF, which is sensitive to reflection [4], and AM, which is mostly viewpoint invariant [4]. Following the approach of Yildirim and colleagues, the neural networks were presented with segmented reconstructions, where non-facial pixels were replaced by a uniform background.

Consistent with previous findings [14], MLMF was more aligned with the CNNs’ mid-level representation, notably the last convolutional layers (Figure 6, A). The AL face patch showed its highest representational alignment with the first fully connected layer (Figure 6, B), coinciding with the surge of the mirror-symmetric viewpoint tuning index at this processing level (see Figure 2). The AM face patch aligned most with the fully connected layers (Figure 6, C).

These correlations between model and neural RDMs reflect the contribution of multiple underlying image features. To disentangle the contribution of reflection-invariant and reflection-sensitive representations to the resulting RDM correlation, we computed two additional model representations for each neural network layer: (1) a reflection-invariant representation, obtained by element-wise addition of two activation tensors, one elicited in response to the original stimuli and the other in response to mirror-reflected versions of the stimuli; and, (2) a reflection-sensitive representation, obtained by element-wise subtraction of these two tensors. The two resulting feature components sum to the original activation tensor; a fully reflection-invariant representation would be entirely accounted for by the first component. For each CNN layer, we obtained the two components and correlated each of them with the unaltered neural RDMs. Through the Shapley value feature attribution method [41], we transformed the resulting correlation coefficients into additive contributions of the reflection-invariant and reflection-sensitive components to the original model-brain RDM correlations (Figure 6, D-F).

In the MLMF face patch, reflection-sensitive features contributed more than reflection-invariant ones, consistent with the dominance of reflection-sensitive information in aligning network layers with MLMF data (Figure 6, D). Conversely, in the AL and AM face patches, reflection-invariant features accounted for nearly all the observed model–brain RDM correlations (Figure 6, E and F). For most of the convolutional layers, the contribution of the reflection-sensitive component to AL or AM alignment was negative—meaning that if the layers’ representations were more reflection-invariant, they could have explained the neural data better.”

(main text, Methods section)

“Measuring brain alignment

To measure the alignment between artificial networks and macaque face patches, we used the face-identities-view (FIV) stimulus set [4], as well as single-unit responses to these stimuli previously recorded from macaque face patches [4]. The FIV stimulus set includes images of 25 identities, each depicted in five views: left-profile, left-half profile, straight (frontal), right-half profile, and right-profile. The original recordings also included views of the head from upward, downward, and rear angles; these views were not analyzed in the current study to maintain comparability with its other analyses, which focused on yaw rotations. We measured the dissimilarity between the representations of each image pair using 1 minus the Pearson correlation and constructed an RDM. To assess the variability of this measurement, we adopted a stimulus-level bootstrap analysis, as outlined in [14]. A bootstrap sample was generated by selecting images with replacement from the FIV image set. From this sample, we calculated both the neural and model RDMs. To prevent spurious positive correlations, any nondiagonal identity pairs resulting from the resampling were removed. Subsequently, we determined the Pearson correlation coefficient between each pair of RDMs. This entire process was repeated across 1,000 bootstrap samples.”

Reviewer #1 (Recommendations for the authors):1. The potential mechanisms underlying mirror-symmetric tuning have been investigated in the past as has been acknowledged and intensively discussed in the paper. However, some of the conclusions made in this paper were already made in previous work, although not explicitly tested. As stated by Leibo et al. "the Hebb-type learning rule generates mirror-symmetric tuning for bilaterally symmetric objects, like faces", suggesting that it will also allow for mirror-symmetric tuning for other object categories. It would be important to acknowledge this.

We respectfully hold a different perspective: the Hebb rule requires training on each object category in order to achieve a view-invariant representation of its exemplars. However, our findings indicate a generalization of mirror-symmetric viewpoint tuning across object categories; the model does not have to be trained on a certain category to acquire a view-invariant representation of its exemplars, provided they exhibit bilateral symmetry. Additionally, as highlighted in the discussion’s limitations section, the model by Leibo et al. exhibits reflection invariance only with respect to the horizontal center of the input image, unlike AL neurons (see Figure 2—figure supplement 8).

Moreover, while Yildirim et al. reports mirror-symmetric viewpoint tuning in VGGFace, they find that this tuning is less present than the view-invariant identity tuning, thereby contradicting neural findings in area AL. It would be beneficial to compare mirror-symmetry with view-invariant coding (as introduced in Figure 1A) in the fully-connected layers of the CNNs in this work, or at least to discuss and acknowledge this potential mismatch of CNNs with neural data.

Please refer to the relevant revision quoted in Point 1.3 above. Moreover, as illustrated in Figures 6 and Figure 6—figure supplement 1-3, CNNs trained on object categorization exhibit similarities to face patches comparable to those in the EIG model. We concur that training on face identification with VGGFace results in a model that does not exhibit AL-like mirror-symmetric viewpoint tuning.

2. The authors propose a mirror-symmetric viewpoint tuning index, which, although innovative, complicates comparison with previous work. This index is based on correlating representational dissimilarity matrices (RDMs) with their flipped versions, a method differing from previous approaches that correlated RDMs with idealized model RDMs corresponding to mirror symmetry. The mirror-symmetric tuning reported by Yildirim et al. for VGG-raw (corresponding to VGGFace in this study; Figure S2) appears significantly lower than the current study (Figure S1). To clarify these differences, an analysis using an idealized model, as in prior studies, or to better motivate the index used in this study would be valuable.

We have added a new paragraph in the Methods section detailing the motivation for introducing a new index. Please refer to the relevant revision in Point 1.1 above.

3. The analysis of reflection equivariance vs. invariance in both trained and untrained CNNs' convolutional layers is informative. However, an illustrative figure would enhance clarity and underscore the distinction between equivariance and invariance.

Following this suggestion, we included a new panel in Figure 1D to augment clarity and emphasize the distinction between equivariance and invariance.

4. Faces seem to behave quite differently from all other object categories tested in this study. The mirror-symmetric viewpoint tuning for faces starts very low but then achieves similar values as other categories in the fully-connected layers. Moreover, the difference between task training and between trained and untrained AlexNet is maximal for faces (Figure 4 and S2). None of these differences are discussed in the paper. Given the critical role of faces for this study, the authors should provide speculations or potential explanations for these observations.5. The study's analysis of training task and dataset raises some questions. Some categories, such as faces, seem to depend strongly on the task/dataset compared to others (e.g. boats). Acknowledging this finding and discussing potential reasons could enhance the manuscript.

We revised the caption of Figure 2—figure supplement 3 to explicitly address this point.

Manuscript changes:

“For face stimuli, there is a unique progression in mirror-symmetric viewpoint tuning: the index is negative for the convolutional layers and it abruptly becomes highly positive when transitioning to the first fully connected layer. The negative indices in the convolutional layers can be attributed to the image-space asymmetry of non-frontal faces; compared to other categories, faces demonstrate pronounced front-back asymmetry, which translates to asymmetric images for all but frontal views (Figure 2—figure supplement 1). The features that drive the highly positive mirror-symmetric viewpoint tuning for faces in the fully connected layers are trainingdependent (Figure 2—figure supplement 2), and hence, may reflect asymmetric image features that do not elicit equivariant maps in low-level representations; for example, consider a profile view of a nose. Note that cars and boats elicit high mirror-symmetric viewpoint tuning indices already in early processing layers. This early mirror-symmetric tuning is independent of training (Figure 2—figure supplement 2), and hence, may be driven by low-level features. Both of these object categories show pronounced quadrilateral symmetry, which translates to symmetric images for both frontal and side views (Figure 2—figure supplement 1).”

6. It is surprising that the face-trained VGGFace shows lower mirror-symmetric viewpoint tuning as VGG16 (Figure S1). What could be possible explanations for this result? Could it be due to an overfitting to natural images of faces (compared to the Basel Face model faces used in this study) of the VGGFace network? A discussion of this finding would be beneficial to the paper.

This point is now addressed explicitly in the caption of Figure 2—figure supplement 5.

Manuscript changes:

“Yildirim and colleagues [14] reported that CNNs trained on faces, notably VGGFace, exhibited lower mirror symmetric viewpoint tuning compared to neural representations in area AL. Consistent with their findings, our results demonstrate that VGGFace, trained on face identification, has a low mirror-symmetric viewpoint tuning index. This is especially notable in comparison to ImageNet-trained models such as VGG16. This difference between VGG16 and VGGFace can be attributed to the distinct characteristics of their training datasets and objective functions. The VGGFace training task consists of mapping frontal face images to identities; this task may exclusively emphasize higher-level physiognomic information. In contrast, training on recognizing objects in natural images may result in a more detailed, view-dependent representation. To test this potential explanation, we measured the average correlation-distance between the fc6 representations of different views of the same face exemplar in VGGFace and VGG16 trained on ImageNet. The average correlation-distance between views is 0.70±0.04 in VGGFace and 0.93±0.04 in VGG16 trained on ImageNet. The converse correlation distance between different exemplars depicted from the same view is 0.84±0.14 in VGGFace and 0.58±0.06 in VGG16 trained on ImageNet. Therefore, as suggested by Yildirim and colleagues, training on face identification alone may result in representations that cannot explain intermediate levels of face processing.”

7. The authors suggest that mirror-symmetric viewpoint tuning is a stepping stone towards view invariance. However, the study lacks explicit testing that this tuning is necessary for viewpoint invariance to emerge. For instance, if one prohibits a network to learn mirror-symmetric viewpoint tuning, would it still achieve viewpoint invariance for bilaterally symmetric objects? While such an analysis might be beyond the scope of this study, it could be beneficial to discuss this as a future direction.

We believe that mirror-symmetric viewpoint tuning is not strictly necessary for achieving view-invariance. However, it is a plausible path from view-dependence to view invariance. We addressed this point in the updated limitations subsection of the discussion.

Manuscript changes:

“A second consequence of the simulation-based nature of this study is that our findings only establish that mirror-symmetric viewpoint tuning is a viable computational means for achieving view invariance; they do not prove it to be a necessary condition. In fact, previous modeling studies [10, 19, 61] have demonstrated that a direct transition from view-specific processing to view invariance is possible. However, in practice, we observe that both CNNs and the face-patch network adopt solutions that include intermediate representations with mirror-symmetric viewpoint tuning.”

Reviewer #2 (Recommendations for the authors):I believe you have all the tools available to address the major weakness with respect to model architectures: https://github.com/CNCLgithub/EIG-faces provides code for the EIG model from Yildirim et al. 2020, and this model also includes pooling and versions of it have been trained for other datasets. As far as I am aware, the EIG model is state-of-the-art for the neural data of mirror-symmetric viewpoint tuning, so applying your analyses to this model would make them a lot more relevant and general.

Thank you for your suggestion. We have integrated the EIG model from Yildirim et al. (2020) into our analysis, using the resources provided in the GitHub link. This has allowed us to extend our comparisons to include state-of-the-art models, enhancing the relevance and scope of our findings. Results are now included in the revised Figure 2—figure supplement 3, and new Figure 2—figure supplement 5 and Figure 6—figure supplement 1-3.

You could also see if more recent "standard" convents such as ResNet-50 explain the neural data at least as well as EIG and then generalize your analyses to this (and other) model(s). To make this case you would have to quantitatively compare the two models on their neural alignment (I believe the data from Yildirim's paper should be public). You might also say that AlexNet and VGG are in the same ballpark for explaining the neural data, but then you should include that analysis in the paper and in the Yildirim paper, VGG seems to fall behind EIG's neural similarity by 10-20 percent points (Figure 3 D iv and E iv).

The findings of Yildirim et al. pertain to VGGFace, a VGG16-like architecture trained for face identification. Unlike VGG16, which is trained on the ImageNet dataset that encompasses a wide array of objects with significant intra- and inter-class variability, the VGGFace network’s training dataset lacks such diversity, consisting primarily of frontal views of faces. Therefore, VGGFace’s weaker brain alignment compared to EIG may stem from its training dataset and task, rather than from its architecture.

We have broadened our analysis to encompass EIG, ResNet50, ConvNeXt, ViT, and HMAX. Our findings indicate that VGGFace, in line with Yildirim’s research, exhibits one of the lowest values in mirror-symmetric viewpoint tuning compared to other models.

Conversely, networks such as ResNet50, VGG, AlexNet, and ConvNeXt demonstrate results comparable to EIG. These findings are detailed in Figures 2—figure supplement 3 and 5.

Reviewer #3 (Recommendations for the authors):– It would be reassuring to know that the object classes have independent measures of symmetry *on which the networks operate*. If the statements about object-class-specific symmetry came after performing the experiments, then I recommend re-writing so that those statements are interpretations.

Implemented. We have changed the order so that the explanations follow the experimental results. This includes the relevant main text paragraph, as well as the relevant figure—both the order of panels and the phrasing of the figure caption.

Manuscript changes:

“Figure 2C displays the average mirror-symmetric viewpoint tuning index for each object category across AlexNet layers. Several categories—faces, chairs, airplanes, tools, and animals—elicited low (below 0.1) or even negative mirror-symmetric viewpoint tuning values throughout the convolutional layers, transitioning to considerably higher (above 0.6) values starting from the first fully connected layer (fc6). In contrast, for fruits and flowers, mirror-symmetric viewpoint tuning was low in both the convolutional and the fully connected layers. For cars and boats, mirror-symmetric viewpoint tuning was notably high already in the shallowest convolutional layer and remained so across the network’s layers. To explain these differences, we quantified the symmetry of the various 3D objects in each category by analyzing their 2D projections (Figure 2—figure supplement 1). We found that all of the categories that show high mirror-symmetric viewpoint tuning index in fully connected but not convolutional layers have a single plane of symmetry. For example, the left and right halves of a human face are reflected versions of each other (Figure 2D). This 3D structure yields symmetric 2D projections only when the object is viewed frontally, thus hindering lower-level mirror-symmetric viewpoint tuning. Cars and boats have two planes of symmetry: in addition to the symmetry between their left and right halves, there is an approximate symmetry between their back and front halves. The quintessential example of such quadrilateral symmetry would be a Volkswagen Beetle viewed from the outside. Such 3D structure enables mirror-symmetric viewpoint tuning even for lower-level representations, such as those in the convolutional layers. Fruits and flowers exhibit radial symmetry but lack discernible symmetry planes, a characteristic that impedes viewpoint tuning altogether.”

– It would be a great contribution to the field if the authors could clarify the relationship between this work and Baek et al.'s. Can they confirm that untrained networks have mirror-tuned units? If that is not replicable, then the "emergence" framing is accurate, and one can exercise appropriate weighing of that other study. This would help the field. If learning just amplifies this symmetry (by strengthening connections of view-dependent units, for example), that is also helpful to learn.

Implemented. We agree with the reviewer that random initialization may result in units that show mirror-symmetric viewpoint tuning for faces in the absence of training. In the revised manuscript, we quantify the occurrence of such units, first reported by Baek et al., in detail, and discuss the relation between Baek et al., 2021 and our work. In brief, our analysis affirms that units with mirror-symmetric viewpoint tuning for faces appear even in untrained CNNs, although we believe their rate is lower than previously reported. Regardless of the question of the exact proportion of such units, we believe it is unequivocal that at the population level, mirror-symmetric viewpoint tuning to faces (and other objects with a single plane of symmetry) is strongly training-dependent.

First, we refer the reviewer to Figure 2—figure supplement 2, which directly demonstrates the effect of training on the population-level mirror symmetric viewpoint tuning.

Note the non-mirror-symmetric reflection invariant tuning profile for faces in the untrained network.

Second, the above-zero horizontal reflection Second, the above-zero horizontal reflection-invariance referred by the reviewer (Figure 3) is distinct from mirror-symmetric viewpoint tuning; the latter requires both reflection-invariance and viewpoint tuning. More importantly, it was measured with respect to all of the object categories grouped together; this includes objects with quadrilateral symmetry, which elicit mirror-symmetric viewpoint tuning even in shallow layers and without training. To clarify the confusion that this grouping might have caused, we repeated the measurement of invariance in fc6, separately for each 3D object category.

Disentangling the contributions of different categories to the reflection-invariance measurements, this analysis underscores the necessity of training for the emergence of mirror-symmetric viewpoint symmetry.

Last, we refer the reviewer to Figure 5—figure supplement 1, which shows that the symmetry of untrained convolutional filters has a narrow, zero-centered distribution. Indeed, the upper limit of this distribution includes filters with a certain degree of symmetry. This level of symmetry, however, becomes the lower limit of the filters’ symmetry distribution following training.

Therefore, we believe that training induces a shift in the tuning of the unit population that is qualitatively distinct from, and not explained by, random-lottery-related mirror-symmetric viewpoint tuned units. In the revised manuscript, we clarify the distinction between mirror-symmetric viewpoint tuning at the population level and the existence of individual units showing pre-training mirror symmetric viewpoint tuning, as shown by Baek et al.

Manuscript changes: (Discussion section)

“Our claim that mirror-symmetric viewpoint tuning is learning-dependent may seem to be in conflict with findings by Baek and colleagues [17]. Their work demonstrated that units with mirror-symmetric viewpoint tuning profile can emerge in randomly initialized networks. Reproducing Baek and colleagues’ analysis, we confirmed that such units occur in untrained networks (Figure 5—figure supplement 3). However, we also identified that the original criterion for mirror-symmetric viewpoint tuning employed in [17] was satisfied by many units with asymmetric tuning profiles (Figures 5—figure supplement 2 and 5—figure supplement 3). Once we applied a stricter criterion, we observed a more than twofold increase in mirror-symmetric units in the first fully connected layer of a trained network compared to untrained networks of the same architecture (Figure 5—figure supplement 4). This finding highlights the critical role of training in the emergence of mirror-symmetric viewpoint tuning in neural networks also at the level of individual units.”

References:

4 Winrich A Freiwald and Doris Y Tsao. Functional compartmentalization and viewpoint generalization within the macaque face-processing system. Science, 330(6005):845–851, 2010. doi:10.1126/science.1194908.

10 Amirhossein Farzmahdi, Karim Rajaei, Masoud Ghodrati, Reza Ebrahimpour, and Seyed-Mahdi Khaligh-Razavi. A specialized face-processing model inspired by the organization of monkey face patches explains several face-specific phenomena observed in humans. Scientific reports, 6 (1):1–17, 2016. doi:10.1038/srep25025.

14 Ilker Yildirim, Mario Belledonne, Winrich Freiwald, and Josh Tenenbaum. Efficient inverse graphics in biological face processing. Science Advances, 6(10):eaax5979, 2020. doi:10.1126/sciadv.aax5979.

17 Seungdae Baek, Min Song, Jaeson Jang, Gwangsu Kim, and Se-Bum Paik. Face detection in untrained deep neural networks. Nature Communications, 12(1):7328, December 2021. doi:10.1038/s41467-021-27606-9.

19 Joel Z Leibo, Qianli Liao, Fabio Anselmi, Winrich A Freiwald, and Tomaso Poggio. View-tolerant face recognition and hebbian learning imply mirror-symmetric neural tuning to head orientation. Current Biology, 27(1):62–67, 2017. doi:10.1016

41 L. S. Shapley. 17. A Value for n-Person Games, pages 307–318. Princeton University Press, Princeton, 1953. ISBN 9781400881970. doi:10.1515/9781400881970-018.

60. Chris Olah, Nick Cammarata, Chelsea Voss, Ludwig Schubert, and Gabriel Goh. Naturally occurring equivariance in neural networks. Distill, 2020. doi:10.23915/distill 00024.004.

61. Joel Z Leibo, Qianli Liao, Fabio Anselmi, and Tomaso Poggio. The invariance hypothesis implies domain-specific regions in visual cortex. PLoS computational biology, 11(10):e1004390, 2015. doi:10.1371/journal.pcbi.1004390.

70 Kaiming He, Xiangyu Zhang, Shaoqing Ren, and Jian Sun. Delving deep into rectifiers: Surpassing human-level performance on imagenet classification. In Proceedings of the IEEE International Conference on Computer Vision (ICCV), December 2015.